# Graphene active sensor arrays for long-term and wireless mapping of wide frequency band epicortical brain activity

R. Garcia-Cortadella[1,9], G. Schwesig[2,9], C. Jeschke[3], X. Illa [4,5], Anna L. Gray[6], S. Savage[6], E. Stamatidou[6], I. Schiessl [7], E. Masvidal-Codina[4,5], K. Kostarelos [1,6], A. Guimerà-Brunet[4,5], A. Sirota[2✉] & J. A. Garrido [1,8✉]

Graphene active sensors have demonstrated promising capabilities for the detection of electrophysiological signals in the brain. Their functional properties, together with their flexibility as well as their expected stability and biocompatibility have raised them as a promising building block for large-scale sensing neural interfaces. However, in order to provide reliable tools for neuroscience and biomedical engineering applications, the maturity of this technology must be thoroughly studied. Here, we evaluate the performance of 64-channel graphene sensor arrays in terms of homogeneity, sensitivity and stability using a wireless, quasi-commercial headstage and demonstrate the biocompatibility of epicortical graphene chronic implants. Furthermore, to illustrate the potential of the technology to detect cortical signals from infra-slow to high-gamma frequency bands, we perform proof-of-concept long-term wireless recording in a freely behaving rodent. Our work demonstrates the maturity of the graphene-based technology, which represents a promising candidate for chronic, wide frequency band neural sensing interfaces.

[1] Catalan Institute of Nanoscience and Nanotechnology (ICN2), CSIC and BIST, Campus UAB, Bellaterra, 08193 Barcelona, Spain. [2] Bernstein Center for Computational Neuroscience Munich, Faculty of Medicine, Ludwig-Maximilians Universität München, Planegg-Martinsried, Germany. [3] Multi Channel Systems (MCS) GmbH, Reutlingen, Germany. [4] Instituto de Microelectrónica de Barcelona, IMB-CNM (CSIC), Esfera UAB, Bellaterra, Spain. [5] Centro de Investigación Biomédica en Red en Bioingeniería, Biomateriales y Nanomedicina (CIBER-BBN), Madrid, Spain. [6] Nanomedicine Lab, National Graphene Institute and Faculty of Biology, Medicine & Health, University of Manchester, Manchester, UK. [7] Division of Neuroscience and Experimental Psychology, School of Biological Sciences, Faculty of Biology, Medicine and Health, University of Manchester, Manchester M13 9PT, UK. [8] ICREA, Pg. Lluís Companys 23, 08010 Barcelona, Spain. [9] These authors contributed equally: R. Garcia-Cortadella, G. Schwesig. ✉email: sirota@biologie.uni-muenchen.de; joseantonio.garrido@icn2.cat

Increasing the bandwidth of neuroelectronic interfaces in terms of spatial resolution and sensitivity in a wide frequency range is a major and ongoing challenge in neural engineering. In the last decades, large efforts have been dedicated to the development of neural sensing interfaces with high sensor-count on conformal substrates[1–10], which are required for highly biocompatible intracranial neural probes[11–14]. In this line, active sensors have emerged as a promising building block for high-bandwidth neural interfaces[4,6,15–19] because they can be arranged in a multiplexed array[2,4,6–9] enabling high sensor-count probes. The detection principle of active sensors is typically based on the modulation of the conductivity of a transistor channel, which is electrically coupled with the biological environment through its gate[8,9,15,18,20–22], producing a local signal pre-amplification. Although active sensing technologies present substantial advantages over conventional micro-electrode arrays, their implementation is currently limited by the demanding material properties required. In order to achieve long-term and highly sensitive neural recordings, materials for active sensing are expected to exhibit semiconducting or semimetallic properties, a high electrical mobility and low intrinsic noise, in addition to a high stability, easy integration in flexible substrates and biocompatibility. Some active sensors based on organic semiconductors and thin Si nanomembranes have exhibited promising performance, with novel transistor architectures[17,22] and insulating technologies[4,14] improving their performance in some typically constrained aspects such as their frequency response or their long-term stability. Graphene-based active sensors are another promising candidate to meet these requirements due to the flexibility of graphene[23,24], its high expected stability[25] and biocompatibility[26,27], as well as its electronic properties, including a high mobility of charge carriers[28,29]. Graphene solution-gated field-effect transistors (g-SGFETs) have demonstrated a high sensitivity for the detection of local field potentials[15] (LFP), as well as a high performance in multiplexed operation[6,7]. In addition, g-SGFETs have recently demonstrated a high sensitivity for the mapping of infra-slow (<0.5 Hz) brain activity (ISA)[30–32] with high spatial resolution[6,7,33,34].

ISA has recently attracted increasing attention due to its unique neurophysiological basis[30] and its relation to resting state networks[31,35–37] and to brain states[36,38–40]. To date, ISA has been typically studied using full-band electroencephalography (fb-EEG)[41,42]. However, increasing spatial resolution of ISA monitoring by using small size electrodes is ultimately limited by the dependence of the amplifier gain on the impedance of the electrodes used. This dependence leads to signal-to-noise loss and signal distortion[43] at low frequencies. For this reason, studies of ISA with high spatial resolution have been typically restricted to indirect measurement methods such as functional magnetic resonance imaging[31,37], optical methods[44] or the analysis of infra-slow changes of signal power at higher frequencies[45]. G-SGFETs, as active sensors, transduce the electrochemical potential signals in the brain ($V_{sig}$) into drain-to-source current ($I_{ds}$) signals (see Fig. 1a). The amplitude of the transduced signals is proportional to the transconductance ($g_m$), defined as the slope of the $I_{ds}$–$V_{gs}$ curves divided by $V_{ds}$ (see Fig. 1b). $g_m$ is proportional to the gate capacitance per unit area (intensive property) and to the $W/L$ ratio of transistor, but not to its active area[17,46–49]. Signal detection based on the field-effect mechanism, therefore, allows to prevent the signal distortion and gain loss observed for small passive sensors in the infra-slow frequency band. This advantage is expected to be valid for all FET-based sensor technologies with stable transfer characteristics, however, experimental proof has been only shown for g-SGFETs, which present a particularly high chemical inertness[25,33]. The properties of g-SGFETs represent a qualitative change in the study of ISA, allowing to explore its

physiological role with an improved spatial resolution. However, in order to advance in the actual application of g-SGFET arrays, several technical aspects remain to be thoroughly evaluated.

In this article, we present a sensing system composed of a flexible 64-channel g-SGFET array and a wireless headstage (Fig. 1c–f and supplementary information S1), which we use to demonstrate the maturity of this technology in terms of long-term and wide frequency band recording capabilities in freely moving animals from a system perspective. First, the focus is placed on the assessment of the in vitro characteristics of the system; including the yield and homogeneity of the graphene sensors, their intrinsic noise and the impact of the data-acquisition (DAQ) system on the sensitivity of these devices. Second, critical aspects for their chronic application in vivo have been resolved; including the stability of the graphene doping, the long-term stability of the g-SGFETs sensitivity and their acute, as well as chronic biocompatibility. Finally, we have applied this methodology to monitor the epicortical local field potentials (LFP) in a freely moving rat model simultaneously with its three-dimensional (3D)-position during long sessions of up to ~24 h. The combination of behavioral and electrophysiological data has been used to assess the capabilities of the wireless recording system to monitor brain dynamics across unperturbed alternation of brain states and validate its sensitivity to detection of high-frequency oscillations associated with sparse behavioral events. As an illustration of unique features of the g-SGFET recording we provide a first case demonstration of infra-slow topographically specific and brain-state invariant pattern associated with high-voltage spindles (HVS). Furthermore, we find changes in infra-slow signal power between slow wave sleep (SWS) and rapid eye movement (REM) sleep and identify the modulation of theta oscillations and sleep spindles by the phase of the DC-signal infra-slow dynamics during REM and SWS, respectively. The results reported here demonstrate that neural probes based on graphene active sensor arrays represent a mature technology, with a high sensitivity, stability and biocompatibility, which allows to chronically map wide frequency band epicortical brain dynamics in freely behaving animals.

## Results

**Homogeneity and sensitivity of graphene active sensor technology.** For the implementation of graphene active sensor arrays as a readily available tool for neuroscientific research, the maturity of large-scale, flexible graphene electronics is critical. Two of the main challenges in the development of these technologies have typically been the production of high-quality single-layer graphene (SLG) and its transfer onto the required substrate. The development of wafer-scale methods to produce SLG has concentrated many efforts and investment in the last decade[50], recently leading to important progress on the growth of graphene by chemical-vapor deposition (CVD)[51]. Here, we show that the quality of commercially available single-layer graphene produced by CVD and transferred on a flexible polymeric substrate (spin-coated on a 4-inch Si wafer) is high enough to enable the fabrication of g-SGFET arrays with a homogeneous good performance both in terms of $g_m$ and electrical low-frequency noise.

Figure 2a shows the boxplot for the $g_m$ of nine neural probes, each of them containing 64 g-SGFETs (with a size of $100 \times 100$ $\mu m^2$ chosen for mesoscale epicortical LFP analysis[10]). These probes were randomly selected from three wafers, all of them processed in independent batches (see "Methods" for fabrication details). It is possible to observe a high homogeneity and yield in terms of $g_m$, with 99% of channels working (defined as transistors having a $g_m$ above 0.7 times the median). The measured median

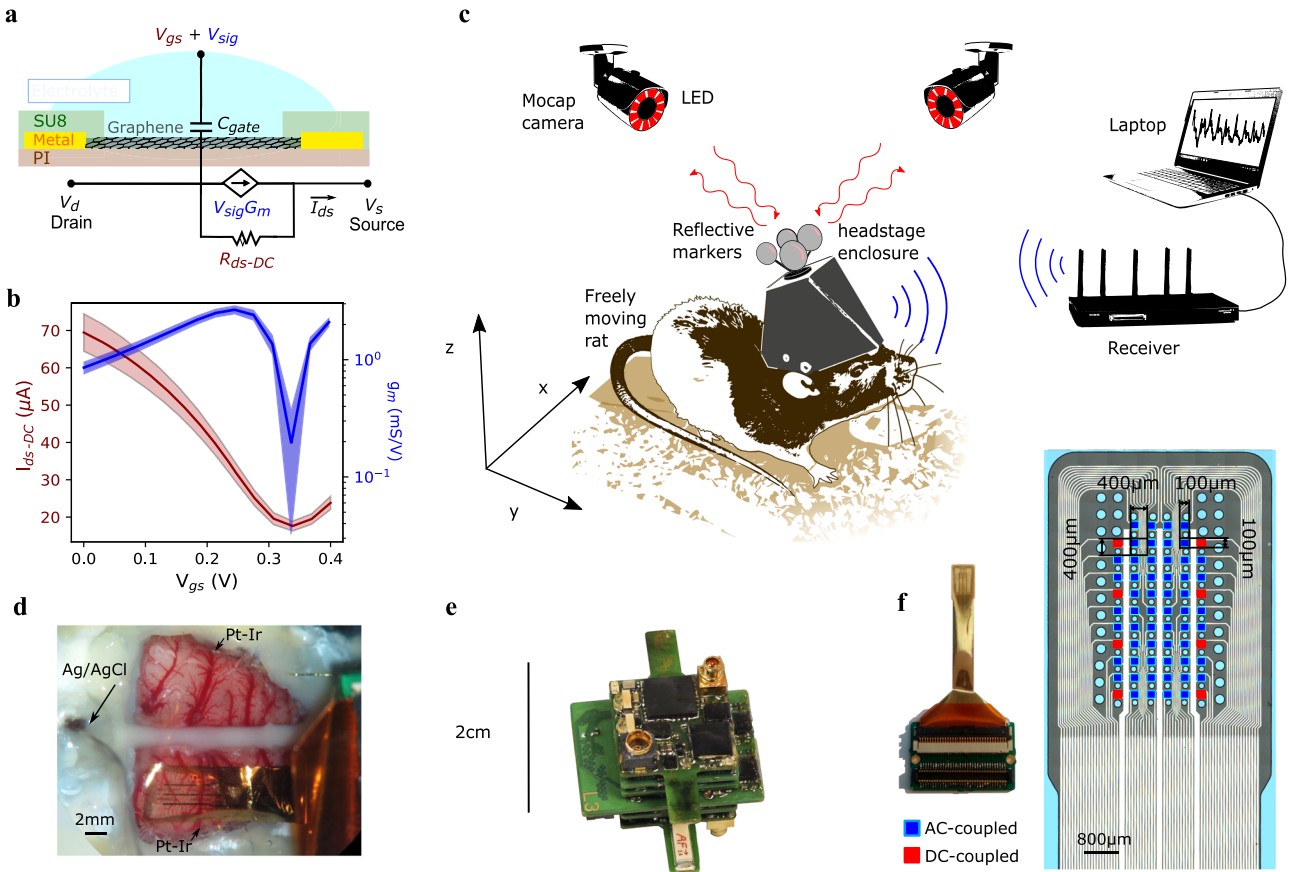

**Fig. 1 Graphene active sensor arrays for chronic, wireless monitoring of wide frequency band epicortical neural activity. a** Schematic of a g-SGFET and its equivalent circuit. The small-signal transduction from voltage to current is modeled by the current source $G_m V_{sig}$, where $G_m \equiv dI_{ds}/dV_{gs}$. The DC current is modeled by the $R_{ds}$ element. **b** Average stationary transfer characteristics of 8 g-SGFETs (left axis) and the $g_m$ of 64 g-SGFETs (right axis). The filled area indicates the standard deviation. **c** Illustration of the rat with the untethered recording system implanted. The headstage and the 3D-printed frame to hold it are covered by a 3D-printed enclosure. On top, the position markers of the motion capture (Mocap) system are fixed, which reflect light back to the Mocap cameras placed in the room. The neural signals transduced by the graphene sensors are digitized and transmitted wirelessly to the signal receiver, which is connected to a computer for signal recording. **d** g-SGFET array placed on the rat cortex; the position of the reference electrode in contact with the cerebellum and two Pt-Ir electrodes at either side of the g-SGFET array are marked with arrows. **e** Photograph of the wireless headstage designed for these experiments. **f** Photograph of the 64 g-SGFET array mounted on a customized connector (left) and zoomed image of the probe active area (right). The red squares indicate the g-SGFETs on the array, which are connected to the headstage inputs with DC capabilities.

$g_m$, 1.9 mS/V, is relatively high with respect to flexible silicon FETs[8] and comparable with typical organic transistor values[17,52] due to the high electrical mobility and gate capacitance of g-SGFETs. In Fig. 2b the equivalent noise at the gate ($V_{gs\text{-rms}}$) of the same devices is shown (see supplementary information S2). $V_{gs\text{-rms}}$ is an important figure of merit to evaluate the sensitivity of the sensors[47], which is defined as the ratio between the integrated current noise ($I_{ds\text{-rms}}$) of the transistor and its transconductance. Although this parameter presents a larger dispersion than $g_m$, it is possible to identify 3 out of 9 probes with 96% of the g-SGFETs showing a $V_{gs\text{-rms}}$ below 10 μVrms, suggesting that the measured noise is not directly related to $g_m$. In fact, low-frequency noise in graphene has been reported to originate from charge trapping-detrapping events[53], which makes noise directly proportional to the density of traps and thus, sensitive to impurities in the environment of graphene. Figures 2c, d show the distribution of $g_m$ and $V_{gs\text{-rms}}$, respectively, for the probe #3 labeled with an asterisk in Fig. 2a. The dispersion in the transconductance of the g-SGFETs can be taken into account in the calibration of the neural signals, correcting dispersion in the signal amplification. Therefore, the truly limiting factor in terms of homogeneity of the g-SGFETs performance is the equivalent noise at the gate. $V_{gs\text{-rms}}$ presents a log-normal distribution[54] with

a mean of 4.13 μVrms and a standard deviation of 1.14 μVrms (excluding the outliers shown in Fig. 2b). These results show that graphene-based neural probes prepared using a 4-inch wafer-scale fabrication process can be obtained with a high homogeneity and sensitivity. Additionally, upscaling of the fabrication process to an industrial scale is expected to further improve the homogeneity of g-SGFETs characteristics, especially in terms of the contamination-dependent charge noise[47,53].

**Wireless headstage design and characteristics.** Another aspect that contributes to the sensitivity of the recording system is the noise introduced by the headstage in the amplification and digitization process. The amplification of the wide frequency band activity requires a DC-coupled system, which implies the digitization of signals with large DC-offsets. In order to digitize signals with such a large dynamic range and minimize quantization noise, a two-stage transimpedance amplifier has been implemented (see schematic in Fig. 2e). The first stage converts the $I_{ds}$ currents from the g-SGFETs into voltage, which contains a wide frequency band signal, including the infra-slow frequency components of $I_{ds}$. In the second amplification stage (see Fig. 2e), the signal is high-pass filtered to remove the DC offset and fill the full scale of the analog-to-digital converter (ADC). In order to

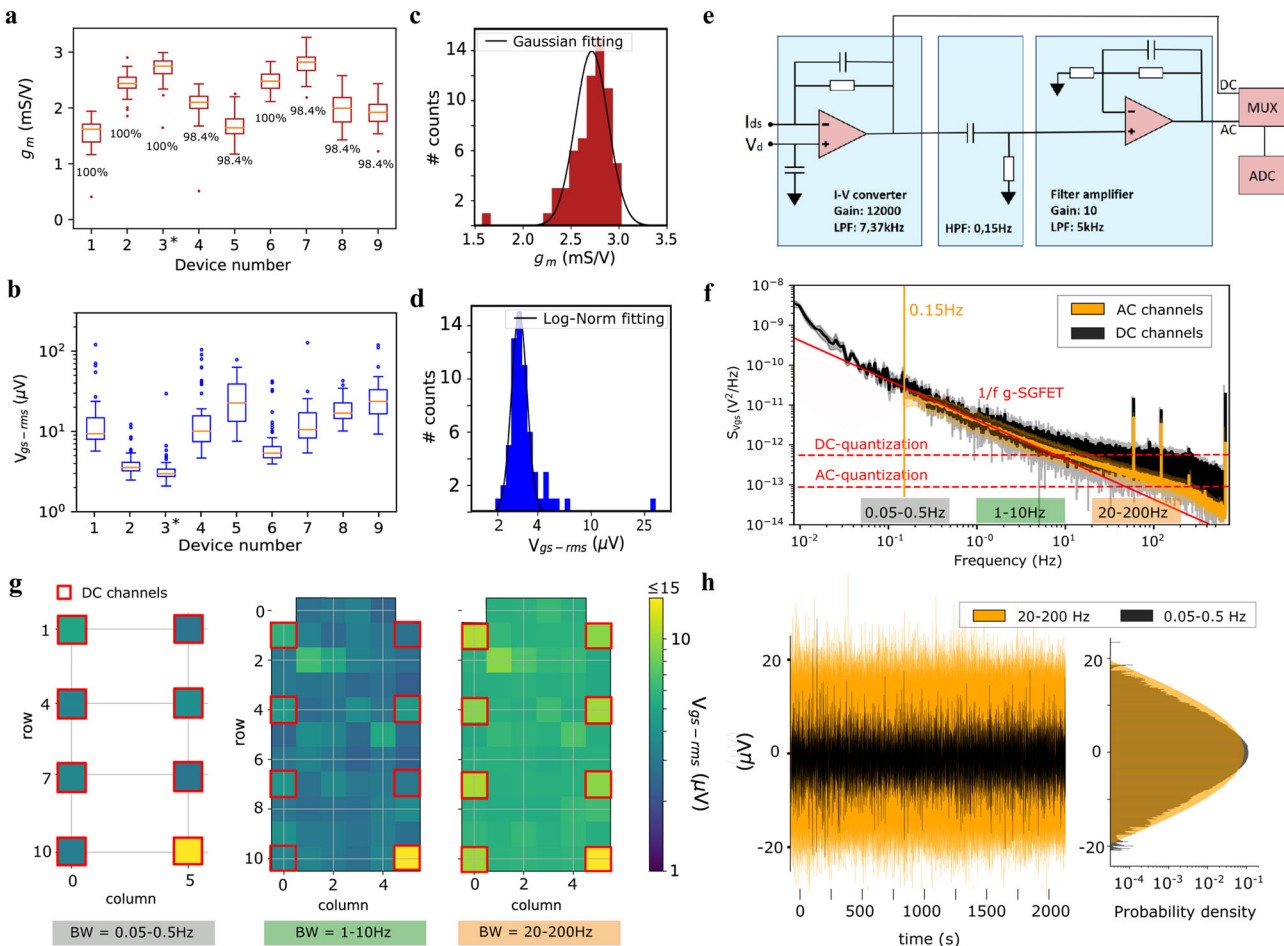

**Fig. 2 Evaluation of system sensitivity. a** Boxplot of $g_m$ for nine randomly selected probes from three different wafers produced in independent batches. The yield in terms of $g_m$ above 70% of the median is indicated. **b** Boxplot for $V_{gs\text{-}rms}$ measured in the 1–10 Hz frequency range, plotted for the same neural probes evaluated in part **a**. All probes consisting of 64 g-SGFETs. The boxes extend from the lower to the upper quartiles, with a line at the median. The whiskers extend 1.5 times the inter-quartile range and the data points beyond the whiskers are indicated by a dot. **c** Histogram of $g_m$ for the 64 g-SGFETs of probe #3 (labeled in panel **a**) and Gaussian fit of the histogram excluding the outliers shown in panel **a**. **d** Histogram of $V_{gs\text{-}rms}$ for the 64 transistors in probe #3 (see panel **b**) and log-normal fit of the histogram excluding the outliers shown in panel **b**. **e** Equivalent circuit of the wireless headstage. **f** Power spectral density (PSD) of the noise from DC channels (black) and AC channels (orange) in probe #3. The 1/f dependence is represented by the solid red line. The vertical orange line indicates the hardware high-pass filter applied to AC channels at 0.15 Hz. The quantization noise of the DC and AC channels is indicated by the horizontal dashed red lines. **g** Representation of the $V_{gs\text{-}rms}$ for all g-SGFETs in probe #3 shown for different bandwidths; 0.05-0.5 Hz band for the DC-channels (left), 1–10 Hz band (middle), and 20–200 Hz (right). The position of the g-SGFETs on the array connected to DC-channels of the headstage is indicated by the red squares. **h** Time domain representation of the noise spectra shown in part **f** and **g** (DC-channels filtered in the 0.05-0.5 Hz band and AC channels in the 20–200 Hz band). Signal from eight channels are overlapped.

dynamically choose between a DC or AC coupling for each channel, a multiplexer has been added to switch between the output of the first and the second stages, see Fig. 2e. Multiplexers have been implemented in only 8 of the 64 channels in order to minimize power consumption and, therefore, battery weight and volume of the dedicated wireless headstage.

Having a relatively high transconductance is important in order to pre-amplify the signals above the noise floor of the transimpedance amplifiers. However, active sensors typically present an intrinsic 1/f noise, which scales with the drain-to-source current[47]. Therefore, $V_{gs\text{-}rms}$ is a more suitable figure of merit to evaluate the sensitivity of active sensors. In order to validate that the sensitivity of the recording system is limited by the intrinsic noise of the active sensors, it is paramount to evaluate the impact of the amplification electronics on the sensitivity of the system in a wide frequency band. The noise level for DC and AC channels can be evaluated from the power spectral density (PSD) of the equivalent voltage noise at the gate

($S_{Vgs}(f)$), defined as the PSD of the current noise over the transconductance (see Fig. 2f). The central part of the spectrum, from roughly 0.05 Hz to 10 Hz, is dominated by the 1/f intrinsic noise of the graphene transistors[53]. For frequencies below 0.05 Hz, the DC-channels show a slight increase above the 1/f noise, which is attributed to the contribution of additional noise sources in the amplification chain, leading to slightly larger $V_{gs\text{-}rms}$ values in the 0.005–0.05 Hz band (see supplementary information S2). Above 10 Hz the noise spectra present a significant increase above the 1/f noise, caused by the quantization noise of the headstage amplifiers, which is more pronounced in DC-channels. The $S_{Vgs}(f)$ integrated in different frequency bands is shown in Fig. 2g for all channels on the neural probe #3. The three maps demonstrate the similarity of the sensitivity of the system in the different ranges, with only a significant increase in the 20–200 Hz band. In this band, the noise of the DC-channels exceeds the noise of the AC channels; however all the graphene sensors (except for an outlier) keep $V_{gs\text{-}rms}$ values below 15 µV.

The digitization noise for AC channels might be decreased by further optimizing the gain of the second amplification stage. However, the intrinsic noise of the amplifier is expected to dominate for large amplification gains. In order to better illustrate the constant sensitivity over frequency, Fig. 2h shows the time domain representation of the noise signal filtered in the ISA band (0.05–0.5 Hz) and the high-frequency band (20–200 Hz). The histogram plotted next to the time-domain representation of both signals shows their probability density distribution, which demonstrates the similarity of their variance, as expected from the integration of a 1/f spectrum in these frequency bands. Note that the apparently lower amplitude in the time-domain representation of the infra-slow noise is due to the different timescales of 1/f noise in both frequency bands, but not due to a different signal variance.

These results show the high sensitivity of the system in a wide frequency band, with $V_{gs-rms}$ below 5 µV in the infra-slow frequency band. In the design of the headstage, we have considered the compromise between reaching maximum sensitivity in the high-frequency range and minimizing the power consumption of the DC-coupled recording system with a relatively high channel count. Smaller g-SGFETs are expected to present a higher intrinsic noise (see supplementary information S2), as expected for any active or passive sensor. Therefore, our results indicate that the sensitivity of g-SGFETs in the infra-slow frequency is not affected by the amplification electronics for sensor areas below $100 \times 100$ µm. This is in strong contrast with ISA detection using passive electrodes, for which the gain loss and signal distortion is expected to increase for smaller sensor dimensions. These results demonstrate the limits and the scalability of the g-SGFET technology towards higher density arrays with ISA detection capabilities.

**Signal stability and sensitivity over time**. Once the performance of the graphene transistors and the headstage is properly assessed, the stability of the g-SGFETs in an in vivo chronic setting needs to be evaluated in order to ensure the reliability of the recording system.

The $I_{ds}$–$V_{gs}$ curves of the g-SGFETs describe the relationship between the measured drain-to-source current and the electrochemical potential at the graphene-electrolyte interface. The minimum in $I_{ds}$ occurs at a particular gate voltage, referred to as the charge neutrality point (CNP), which is also related to a minimum in the sensitivity of the device (see Fig. 1a). The CNP corresponds to the bias conditions for which the Fermi energy in the graphene channel is, on average, closest to the energy with a minimum density of states (i.e., the Dirac point)[55]. The $V_{gs}$ overpotential required to reach this energy depends on the doping[56] of the graphene channel, as well as on the electrochemical potential of the reference electrode. Therefore, instabilities in any of these two parameters will produce a shift of the transfer characteristics in the $V_{gs}$ axis. In turn, this shift implies that $I_{ds}$ will present a drift and that the sensitivity of the g-SGFETs might vary over time for a constant $V_{gs}$ overpotential. Having a controllable doping of the g-SGFET and a homogeneous CNP among sensors is, therefore, of paramount importance to maintain a good sensitivity of the sensor array.

Figure 3a shows the evolution of the transfer characteristics over 4 weeks after implantation of the neural probe (see "Methods" for implantation details). The observed shift in the CNP is presumably due to a combination of factors, including desorption of contaminants by electrochemical cleaning of the graphene-electrolyte interface[57], adsorption of charged chemical species present in the environment or changes in the reference electrode potential (see supplementary

information S3). However, from these results it is not possible to distinguish among all different contributions. The accumulated drift in the CNP measured during the first 24 h of recording reaches approximately 50 mV, with a maximum change rate of ∼20 mV/h in the first hour (see supplementary information S3). Figure 3b shows the measured signal in two DC-coupled channels (high-pass filtered at 1 mHz) during the first 2 h of recording. Figure 3c shows the amplitude-phase relationship between these two DC-coupled channels in the 0.005–0.05 Hz band (see "Methods" section). The left panel shows the amplitude-phase coupling measured in PBS, while the right panel shows the equivalent results in vivo. The in vivo signals exhibit fluctuations with a much larger amplitude than the signals recorded in PBS, ruling out the transistor 1/f noise and the headstage noise as the origin of these infra-slow oscillations. Further, the in vivo signals recorded in the 0.005–0.05 Hz band present fluctuations in anti-phase, which confirm that neither instabilities in the reference electrode nor adsorption/desorption of chemical species on graphene are responsible for these fluctuations. To conclude this discussion, Fig. 3d shows the effect of drifts in the graphene doping on the $V_{gs-rms}$ of graphene sensors. These results demonstrate that their sensitivity does not change significantly due to the accumulated drifts during up to 24 h if the initial bias is selected properly. Therefore, daily tracking of the CNP and readjustment of the $V_{gs}$ overpotential back to optimum values is enough to keep a constant sensitivity over long-term monitoring of the brain dynamics.

In addition to changes in the doping of graphene, the transconductance and noise of the g-SGFETs might vary over time due, for instance, to the creation of defects in the graphene lattice. Pristine graphene has shown excellent chemical stability due to its sp2 hybridization[25]. However, dangling bonds at edges, grain boundaries, atomic vacancies or reconstructions in the atomic lattice increase the reactivity of graphene, which might lead to the creation of defects over time[25]. In addition, there might be mechanical causes of performance degradation such as detachment of graphene from the substrate or bending-induced strain on the graphene lattice and metal-graphene contacts. Another possible cause of sensitivity degradation could be the encapsulation of the device by glial scar tissue[58]. This layer of tissue can be modeled as an electrical impedance in series with the graphene-electrolyte interface[59], which can eventually lead to a degraded frequency response of the g-SGFETs.

In order to track changes in the sensitivity over time in a chronic implant, the $g_m$ extracted from the $I_{ds}$–$V_{gs}$ curves and the $V_{gs-rms}$ were periodically measured for the 8 DC-coupled channels over 4 weeks. Figure 3e shows that $g_m$ remained approximately constant, suggesting that there are no major creation of defects in the graphene channel in the in vivo environment. Similarly, the $V_{gs-rms}$ shows only a slight increase in the last days. Figure 3f shows the current noise ($I_{ds-rms}$) for all the 64 channels measured at 200 Hz over 4 weeks. At this frequency, it is possible to estimate changes in the sensitivity of the recording system due to the low average power of high-frequency neural signals (see supplementary information S4). The numeric values displayed in Fig. 3f indicate the percentage of g-SGFETs working (see supplementary information S4). The frequency response of the transconductance ($g_m(f)$) has also been measured in vivo over 4 weeks after implantation. For this purpose, two Pt-Ir electrodes were implanted on both sides of the g-SGFET array (see inset in Fig. 3g) and 1 µA amplitude pure tone signals of different frequencies were applied using a current source. Figure 3g shows the magnitude of $g_m(f)$ for different days after implantation of the neural probe normalized by the mean magnitude at 1 Hz; the phase of $g_m(f)$ is shown in the supplementary information S5. The

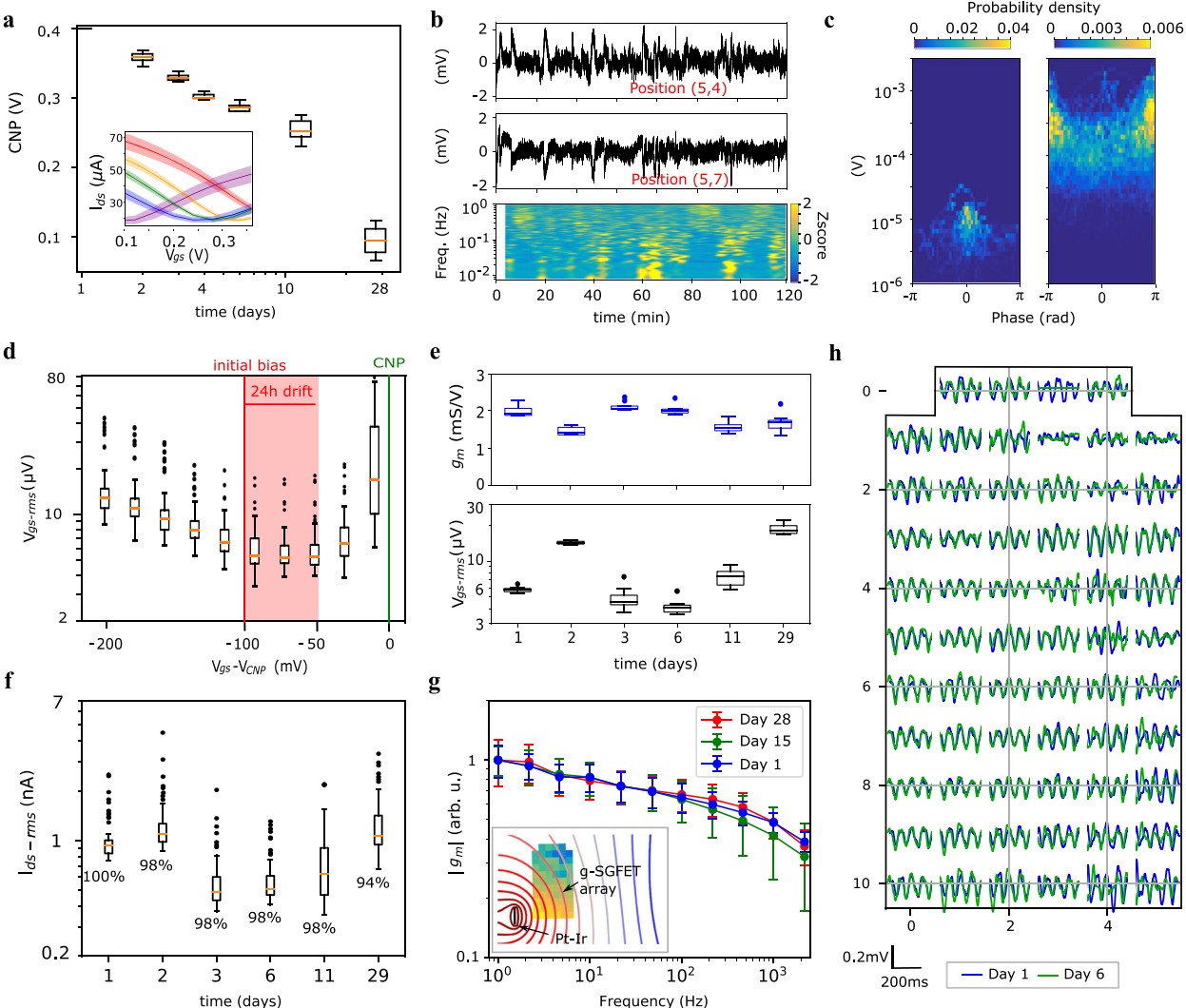

**Fig. 3 g-SGFET stability in-vivo. a** CNP vs. time over 4 weeks. The inset shows the $I_{ds}$–$V_{gs}$ curves. Mean and standard deviation for $n = 8$ g-SGFETs (1 outlier excluded). **b** Signal from two DC-coupled channels. Indicated positions corresponding to map in panel **h**. The spectrogram of channel (5,7) is shown (bottom). **c** The phase-amplitude relation between the channels in panel **b**, for the noise measured in the beaker (left) and for the signals measured in vivo (right). **d** Boxplot of $V_{gs-rms}$ vs. shifts in the effective gating ($V_{gs}$–$V_{CNP}$) of the 64 g-SGFETs. The colored area indicates the measured drift in the CNP referred to a Ag/AgCl electrode during the first 24 h of recording. The initial bias and the CNP are indicated by the red and green vertical lines respectively. **e** $g_m$ (top) and $V_{gs-rms}$ (bottom) measured over 4 weeks post implantation; $g_m$ was obtained from the $I_{ds}$–$V_{gs}$ curves of the DC-coupled channels ($n = 8$ g-SGFETs, 1 outlier excluded). **f** Current noise over 4 weeks after implantation ($n = 64$ g-SGFETs). The numeric values indicate the yield of working devices (see supplementary information S4). The boxes in panels **a** and **d**–**f** extend from the lower to the upper quartiles, with a line at the median. The whiskers extend 1.5 times the inter-quartile range and the data points beyond the whiskers are indicated by a dot. **g** Average and standard deviation of the frequency-dependent transconductance ($|g_m|$ ($f$)) shown for different days after the implantation ($n = 10$ g-SGFETs). The inset shows the approximate position of the Pt-Ir electrode close to the array, the simulated equipotential contour lines in a conductive plane and the relative signal amplitude measured by each of the g-SGFETs in the array (see supplementary information S5). **h** Signals measured by all g-SGFETs on the array during a state of increased theta activity on day 1 and day 6 after implantation.

approximately constant slope (in a log-log scale) follows a fractional order attenuation (i.e., approximately $\propto 1/f^{0.1}$), which has been recently attributed to the non-ideal capacitive response of the graphene-electrolyte interface[34]. A calibration method to correct such transconductance variation has also been proposed[34]. The evolution of the frequency response does not show major changes in the slope of the $g_m$ attenuation, indicating that there is not a significant increase in the electrical impedance in series with the graphene-electrolyte interface due to device encapsulation[60]. To conclude, Fig. 3g shows the recorded neural activity in a state of increased theta power in day 1 and day 6 after implantation, illustrating the good homogeneity and stability of the g-SGFETs performance. Future studies could address in

greater detail, by studying a large animal cohort, the stability of the biological signal over time, a critical aspect in electrophysiology research and for the long-term performance of brain-computer interfaces[61,62]. Furthermore, the polymers used as a substrate and passivation layers could be modified to reduce the moisture absorption[14,63] and displace the neutral plane of the device at the position of the graphene channel (see "Methods" section). Yet, the results presented in this section reveal a promisingly stable performance over time, which sets a lower bound for the stability of g-SGFETs in a chronic implant environment. Besides, from a system perspective, we show that g-SGFET arrays can measure very slow biological signals (high-pass filtered above 1 mHz).

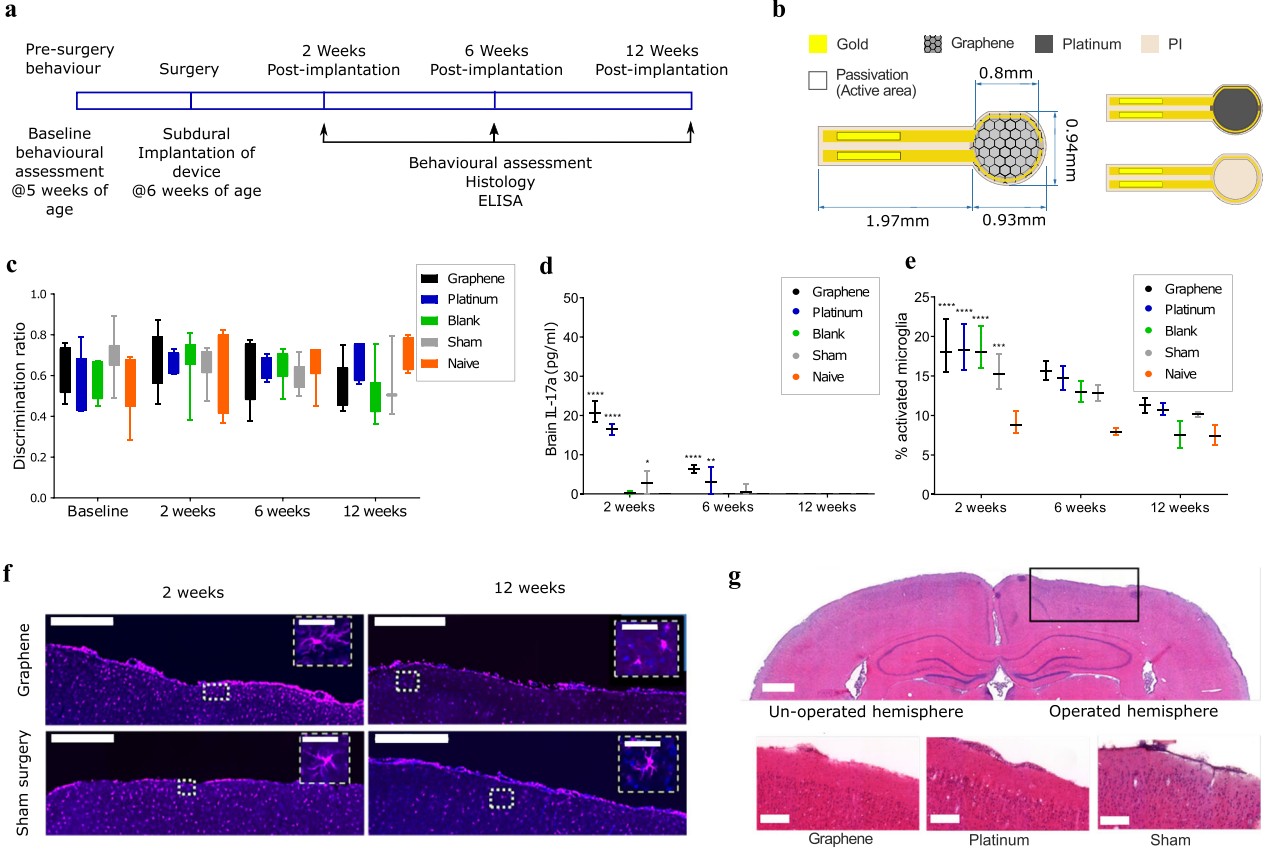

**Fig. 4 Biocompatibility testing of non-functional g-SGFET vs. control devices. a** The timeline describes the procedures carried out on animals during the biocompatibility study. **b** Schematic of the high-surface area g-SGFET prototype developed for biocompatibility testing in-vivo. **c** Discrimination ratio from NOR test over different days after implantation (see "Methods" section). For all five groups tested, the discrimination ratio was above 0.5 at all timepoints. Evaluated for $n = 7$ animals per group at all timepoints, except 12 weeks which had $n = 3$ (sham), $n = 4$ (platinum and naive) and $n = 7$ (blank). The boxes from the lower to the upper quartile, while whiskers represent minimum and maximum values. **d** Inflammatory marker IL-17a in the brain tissue for all groups and timepoints. Evaluated for $n = 4$ animals after 2 and 12 weeks, and $n = 3$ animals at 12 weeks. **e** Microglial activation state, expressed as a percentage of total microglial presence in the site surrounding the electrodes. $n = 3$ animals at 2 and 12 weeks, $n = 2$ animals (or 3 for the contralateral hemisphere) at 6 weeks. Bars in panels **d** and **e** indicate the mean and range of data point. **f** Iba-1 immunoflourescent staining to assess activation status of microglia at the surgical site obtained from 40 sections per animal. Scale bar equals 500 μm (50 μm at the insets). **g** Haemotoxylin and Eosin staining at 2 weeks post implantation shows there is no structural damage to the cortical layers directly at the device implantation site. Forty sections at 25 μm per animal were imaged. Scale bar equals 1 mm (top) and 200 μm (bottom). In panels **d** and **e** two-way ANOVA test with Dunnett's multiple comparison to the naive control within each timepoint with $n = 3$ or larger: *, **, ***, and **** indicate $p = 0.015$, $p = 0.007$, $p = 0.0016$, and $p < 0.0001$, respectively.

## Biocompatibility of graphene devices following subacute and chronic implantation

In order to assess the applicability of g-SGFET arrays for the long-term monitoring of brain activity under natural behavior we have also investigated the biocompatibility of graphene-based epicortical devices. For this purpose, animals were implanted with one of three devices onto the parietal cortex of the brain, or had the full surgery without the implantation of any device (sham control). A cohort of naive animals who had no intervention were used as a control. Three time points were chosen to assess tissue response: 2 weeks, 6 weeks, and 12 weeks post implantation (Fig. 4a). Non-functional devices were custom-designed with an enlarged surface area of CVD graphene, in order to maximize exposure of the material to the brain tissue (see Fig. 4b for device dimensions). The experiments were designed following the guidance from the ISO 10993 standard, which details the biological evaluation of medical devices. Ethylene oxide sterilization was applied prior to implantation[64]. After implantation, the immunohistochemical response of the tissue and potential effects on the behavior were investigated.

Behavior was assessed with the novel object recognition (NOR) test, used to assess impairment of cognition and memory[65]. No significant differences in the discrimination ratio were found in animals implanted with any device at any of the timepoints (Fig. 4c). The inflammatory response of the tissue was evaluated using two main techniques: ELISA of blood or brain tissue for a panel of inflammatory cytokines, and immunohistochemical analysis of brain tissue for cells associated with inflammation. ELISA was performed for four cytokines: interleukin-6 (IL-6), interleukin-17a (IL-17a), interferon gamma (IFN-γ), and tumor necrosis factor alpha (TNF-a). In blood serum, there were no significant differences between any of the materials at any timepoints (see Fig. S6). For cytokine expression in brain tissue, significantly higher levels of all four cytokines in both graphene and platinum devices were found at the 2 weeks timepoint, when compared with the contralateral hemisphere control. Whilst release of these factors is typically intended to prevent further damage to the CNS tissue, prolonged expression can be detrimental. By 6 weeks post implantation, there was still a significant elevation of both IL17a and IFN-γ for graphene and platinum devices vs. control expression, and by week 12, there was no significant expression of any cytokine for any treatment group (Fig. 4d and Fig. S7). These findings demonstrate that the

adverse tissue response to graphene is transient in nature, comparable to the current clinical standard and specific to the implantation site, with no observed systemic complications.

To confirm the ELISA data, manual counting of the activation state of microglial cells was also performed to assess the inflammatory state within the brain. Microglial cells are always present within the brain, but their morphology serves as an indicator of the inflammatory state within the brain[62]. Expression of activated microglia was increased at both the 2 weeks and 6 weeks post implantation timepoints, and this activation was present to a significant level at 2 weeks post implantation in all four treatment groups when compared with the contralateral hemisphere. However, similar to the ELISA, the activation of microglia had returned to baseline levels by 12 weeks, indicating no prolonged inflammatory reaction to the devices (Fig. 4e, f and Fig. S8). TUNEL cell counting was also performed, to assess any cell death within the tissue as a result of the implantation of devices. At 2 weeks post implantation, there was a significant increase in the number of TUNEL-positive cells for both graphene and sham surgery groups. However, by 6 weeks there was no evidence of cell death, which was also true at 12 weeks (Fig. S9). Finally, there was no obvious morphological changes seen with haemotoxylin and eosin staining. There was an appearance of sunken cortex in some brains, however, this was due to perfusion fixation with the glass window in place, and there was no effect on the thickness of the cortical layers below the implantation site, as shown in Fig. 4g.

Overall, both cytokine expression and histological analysis of the brain area at the implantation site showed an acute reaction to the implantation of devices. However, this was not specific to the graphene devices, even though an enlarged surface area of graphene was used in order to maximize the material specific response. By 6 weeks, the reaction showed clear signs of amelioration, and by 12 weeks, there was no detectable reaction to the devices using any technique. Graphene and Pt devices showed a similar level of microglia activation compared to "blank" devices, while the latter shows much smaller presence of inflammatory markers than for Graphene or Pt. These results suggest that microglia activation is more strongly associated with surgical procedure and probe insertion, while inflammation is primarily affected by the device material. In this way, functional sensor arrays, which present a much lower graphene area, are expected to cause an inflammation closer to that caused by "blank" devices. In addition, according to NOR test, graphene devices did not affect significantly the animal behavior neither in acute nor chronic timepoints. Based on these results, graphene-based devices presented adequate biocompatibility for chronic implantation, comparable to the equivalent platinum-based devices.

**Long-term monitoring of wide frequency band epicortical brain activity during natural behavior.** During the longitudinal in vivo assessment of the g-SGFET sensitivity, we recorded epicortical brain activity in a freely behaving rat for up to 24 h. Throughout the recording period, the 3D-motion of the animal was tracked with a motion capture (Mocap) system (see "Methods" section and Fig. 5a). The conjunctive recording of animal motion and wide frequency band epicortical signals was used to classify brain states and behavioral states over the recording period. In turn, this classification was used to support two main purposes. First, to validate the ability of the graphene-based wireless recording system to perform long-term stable recordings in freely moving rat across multiple brain states, and test its suitability to study infra-slow epicortical LFP dynamics. Second, to assess the g-SGFET sensitivity in the high-frequency range of

the LFP dynamics related to spontaneous behavior. The analysis of the relationships between epicortical brain activity and freely moving behavior was performed over timescales enabled by the wireless recording system. This capability is critical for the study of sparsely occurring behavior events, as well as ISA patterns over distinct brain states.

Brain states were classified through a combination of spectral features in the epicortical LFP signal and motor data. In this way we distinguished the following classes: slow wave sleep (SWS), REM sleep (REM), Awake Theta (AwT), and Awake Non-Theta (AwNT). Figure 5b illustrates the criteria for the brain states classification, described in detail in the "Methods" section. First, slow wave (SW) states, showing increased power in the 1–25 Hz band, and Theta states were identified (see Fig. 5b). The behavior of the animal was then classified in either active or inactive periods from the motion tracking data (see Fig. 5b and "Methods" section). During inactive motor behavior, SW states were classified as SWS except in direct proximity to HVS events, while Theta states were categorized as REM if directly preceded by SWS. On the other hand, Theta states occurring during active behavior were classified as AwT and SW states as AwNT. Finally, periods not assigned to Theta or SW states were classified as AwNT regardless of the behavior of the animal. During the majority of recording hours all four sleep/wake states were expressed at least once in line with the polyphasic nature of rat sleep[66–68]. Their relative prominence however varied substantially over the course of the recording day paralleling the changes observed in motor states (Fig. 5c), in line with circadian rhythmicity.

Classification of brain states is typically based on the delta, alpha-beta and theta frequency bands (see "Methods" section), reflecting fast-time scale state-specific network dynamics. However, some recent research highlighted the role of infra-slow dynamics in the regulation of brain sub-states[40], via modulation of higher LFP frequency bands during sleep[39,45,69] and dynamic coordination and segregation of the resting state[35,70]. These results show the potential importance of ISA for a complete classification and study of brain-states. The graphene-based recording system presented here represents an ideal tool for the study of cortical ISA signals with a high accuracy and spatial resolution in freely behaving animals. The spectrogram in Fig. 5b illustrates changes of the spectral power for frequencies between 0.015 and 4 Hz over the transition between SWS and REM. It is possible to observe clear increase in the ISA-band power following the transition from SWS to REM, even at the single trial level (see Fig. 5b). Taking advantage of the long-term recording capabilities of our system, we could sample 44 of such sparsely occurring SWS-REM (REM duration longer than 40 s) state transitions within a 24 h period. Besides, the spatial mapping of ISA enabled by the g-SGFET technology allows to resolve the topographic region-specific modulation of ISA at the SWS-REM state transition (see supporting information S12). Interestingly, delta-band power, associated with slow oscillations, and infra-slow power showed changes in opposite directions between SWS and REM sleep. While delta band power expectedly decreases from SWS to REM, associated with desynchronized cortical state, infra-slow power increases in REM (see Figs. 5d, e and Fig. S12 and statistical analysis in "Methods").

In order to further illustrate the wide frequency band sensitivity of the recording system, we quantified the strength of modulation of LFP power in the slow frequency range (1–15 Hz) by the phase of the ISA activity during REM and SWS. Interestingly, ISA phase significantly modulated theta power (8–9 Hz) during REM sleep (Fig. 5f-left) and spindle band power (9–13 Hz) during SWS (Fig. 5f-right). The strength of ISA phase modulation was tenfold higher during REM compared

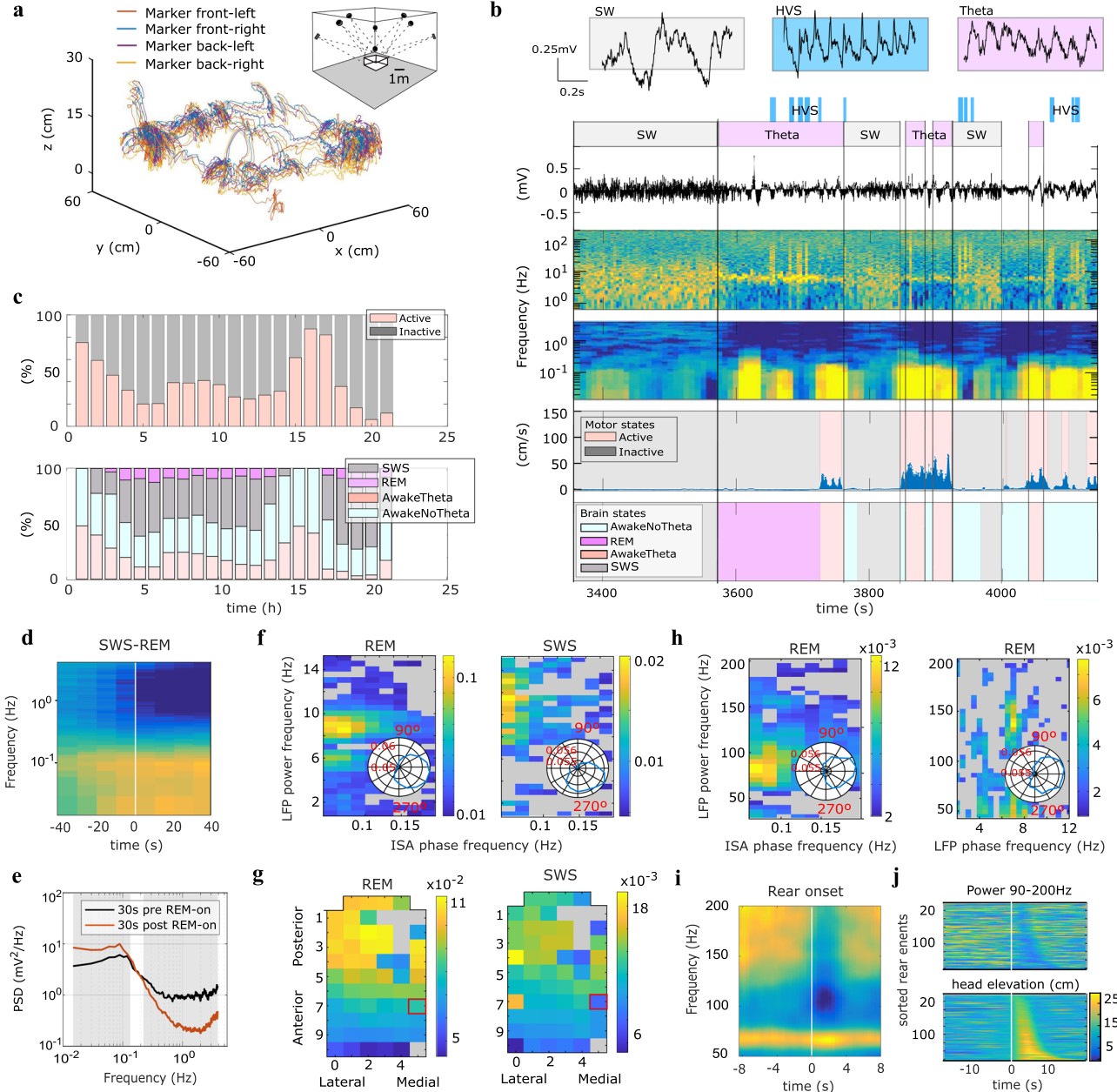

**Fig. 5 Infra-slow to high-gamma correlates of sleep and behavioral states. a** 3D trajectories of the head position of the rat. The inset shows a scheme of the position of the Mocap. **b** The spectrogram and raw LFP signal of an illustrative channel is displayed for distinct brain states (top); slow-wave (SW), high-voltage spindles (HVS) and Theta. Movement speed is displayed along with classification of motor state (middle) and brain states (bottom). **c** Top: percentage of time in the active vs. inactive state (interruptions to replace the battery not included). Bottom: percentage of time the rat was in each main brain state. **d** Average 0.015–4 Hz spectrogram for one DC channel triggered on REM episode onsets ($n = 44$). **e** Median PSD across SWS-REM transition episodes ($n = 44$) for 30 s periods pre and post REM onset. Shaded area marks frequency bins with significant difference ($p < 0.05$, permutation test). **f** Color-coded strength of modulation of LFP power across slow frequency range (y-axis) for one channel by the phase of ISA across 0.05–0.2 Hz range derived from one DC channel (see panel **g**) during REM sleep (left) and SWS (right) states. Gray color indicates nonsignificant modulation. Insets show circular plot of LFP power in theta/spindle band as a function of ISA phase. **g** Color-coded topographic maps of ISA phase modulation of LFP power in theta band during REM (left) and spindle band during SWS (right). ISA phase derived from DC channel marked with a red square. **h** Color-coded strength of modulation of LFP power across gamma frequency range (y-axis) for one channel by the phase of ISA across 0.05–0.2 Hz range derived from one DC channel (left) and by the phase of LFP in the slow frequency range rhythm (right) during REM. Inset, circular plot of LFP gamma power with respect to respective (ISA or theta) phase. **i** Average spectrogram for high-frequency range of LFP on posterior channel triggered on rear onset ($n = 162$). **j** Head elevation (bottom) and high gamma power (top, same channel as in **i**) color-coded and centered on rearing onset shown for all events sorted by duration of rear event.

to SWS, and the ISA phase of maximal LFP power differed between states being close to the peak (~340°) in REM and ascending phase (~300°) in SWS. Taking advantage of the coverage of a significant section of dorsal cortical mantle by our array, we assessed the spatial extent of the ISA phase modulation

of LFP power across cortex, with both theta power during REM and spindle power during SWS showing strongest modulation in posterior part of the array (Fig. 5g). While theta oscillations measured on the cortical surface are generated by volume conduction of multiple theta-rhythmic current generators of

entorhino-hippocampal circuits[71,72], sleep spindles are generated by rhythmic currents of thalamo-cortical projections to granular cortical layers[73]. The fact that power of hippocampal theta and cortical spindle band is modulated by the phase of ISA derived from cortical surface likely reflects global infra-slow dynamics that co-modulates both limbic and cortical circuits. While the topographic profile of theta power (Fig. 5g) modulation by ISA phase is consistent with anatomical localization of underlying hippocampal theta current generators, stronger modulation of the spindle power on posterior cortical areas might reflect anatomical thalamo-cortical subcircuits that are more strongly co-modulated by ISA dynamics than derived from epicortical DC signal. Finally, we tested whether g-SGFETs SNR is sufficient to detect fluctuations in the high-frequency LFP dynamics at different time scales and to this end quantified the strength of modulation of broad range gamma power (30–200 Hz) by both ISA phase and theta rhythm phase during REM sleep. Gamma power in in the range of 60-120 Hz was modulated by the ISA phase reaching maximum power at the peak of the ISA (~10°) (Fig. 5h-left) and, consistently with published work based on intracranial recordings[73], high gamma (120–150 Hz) power was modulated by theta phase (Fig. 5h-right).

Having established that we can record state-selective epicortical signals with g-SGFETs across a range of brain/motor states, we subsequently went on to demonstrate applicability of the technique for linking behavior and cortical physiology. To this end we focused on a specific and sparsely occurring spontaneous behavior, rearing on the hindlimbs. Rearing is a exploratory behavior in rodents, which is context- and stress-sensitive[74,75], has been hypothesized to support sampling of distal landmarks for construction of a cognitive model of the surrounding environment[75,76] and is implicated in modulation of cortico-hippocampal interactions in theta and gamma frequencies[76,77]. In general, due to the sporadic and spontaneous occurrence of rearing events their neural physiology has been less widely investigated with conventional recording methods compared to task-specific trained motor actions. Technologies that combine long-term recording stability, high spatial resolution, wireless methodology and precise 3D-tracking of animal behavior, as presented here, open the door to investigating this class of phenomena with a great level of detail. Therefore, we took advantage of the presented technology to collect a large number of individual spontaneous rearing events during a full 24 h period. Evaluating the signature of rearing on the gamma epicortical activity band is of additional interest for our study, since it can be used to illustrate the capabilities of the g-SGFETs in the high-frequency LFP range. In order to robustly detect rearing episodes we took advantage of the continuous 3D tracking, detecting rearing events ($n = 163$) based on head elevation above ground (see "Methods" and supplementary information S6). While rearing events occurred throughout the recording period, expression of rearing activity was highly variable across the day, as with overall motor activity, ranging from 250 s to 0 s spent rearing per hour (mean $43.8 \pm 12.1$ s, supplementary information S10). Additionally, rearing events expressed variability in terms of height (mean $250,6 \pm 2.7$ mm, see Fig. 5j and Fig. S10d) and duration (mean $5.7 \pm 2.8$ s, see Fig. S10e).

Having detected this set of spontaneous rearing episodes we proceeded to analyze the power spectra of epicortical LFP, which showed distinct rearing-associated changes in brain signals for specific frequency ranges. Rearing was associated with the suppression of epicortical high-frequency (90–200 Hz) activity (Fig. 5i), which was also observable on the single trial level across the range of rear heights (Fig. 5j) and most prominently observed on more frontal channels of our array (see supplementary information S11). In strong contrast no such suppression was observed in the gamma band between 60 and 70 Hz (see Fig. 5i and supplementary information S11).

Subsequently, we took advantage of the infra-slow recording capability of g-SGFET arrays to characterize topographic infra-slow and spectral AC epicortical signals associated with rare highly synchronous high-voltage spindle (HVS) oscillations[78,79], as the most likely cortical dynamics associated with large infra-slow currents, similar to those shown during epileptiform activity in development[80]. Consistent with previous studies[78,79], HVS occurrence was associated primarily with alert immobility states (IMM) (566 events, Fig. 6a), where IMM is defined as the intersection between inactive and awake states. Benefiting from the long-term unperturbed recording allowed by our system we could also sample significant number of HVS events during REM sleep (92 events), where they coexisted with hippocampal theta oscillations visible on posterior derivations (Fig. 6b). While median duration of HVS events was comparable in immobility and REM sleep (~5 s, Fig. 6c-top), the rate of detected HVS events varied across the recorded 24 h period, (Fig. 6c-bottom). Interestingly, HVS during both brain states were associated with transient infra-slow fluctuations as visible in single examples (Fig. 6a, b) and average profiles (Fig. 6d, e). Specifically, positive (putative source) and negative (putative sink) infra-slow transients in, respectively, posterior and frontal positions on the array coincided in duration (median 5 s) with oscillatory dynamics of HVS (Fig. 6c, d, e). Topographic profiles of spectral peak power of HVS were comparable for both states and showed maximal increase to baseline in frontal derivations overlaying sensory-motor cortex (Fig. 6f). REM-associated HVSs were, on average, slower and larger in power than immobility-associated ones (Wilcoxon ranksum test, $p < 1e-19$ for power and $p < 1e-8$ for frequency, Fig. 6g). In contrast, spatial structure and magnitude of infra-slow fluctuations associated with HVS as expressed by positive fluctuations on posterior and negative fluctuation on frontal DC channels, were comparable for both states (Wilcoxon ranksum test between IMM and REM, $p = 0.5$ for posterior peak magnitude and $p = 0.9$ for frontal trough magnitude, Fig. 6h).

Although future work is required to replicate these observations in a large cohort of animals, our long-term recordings enabled detailed quantitative analysis of rare physiological patterns and illustrate the power of this technology. Because of their sparse, strongly state dependent occurrence and distribution over a wide frequency band these events illustrate the class of phenomena whose functional study necessitates the integration of neural-behavioral measurements with both high spatio-temporal resolution and large spatio-temporal span in freely moving animals, as enabled by our wireless electrophysiology system.

## Discussion

Graphene active-sensor arrays represent an emerging technology in neural engineering, which has recently demonstrated a strong potential for the production of high-count sensor arrays[6,7], as well as for wide frequency band neural sensing[33]. In this article, we have presented a detailed characterization of various technical aspects required for their actual application, such as the homogeneity in the performance of the graphene sensors, the specifications required for a dedicated headstage, the limits in the sensitivity of g-SGFETs or their chronic stability and biocompatibility in vivo and demonstrated lines of investigation enabled by their technical characteristics.

In the first place, we have demonstrated the high yield and homogeneity of the g-SGFETs produced in a wafer-scale process using commercially available CVD graphene. This demonstration

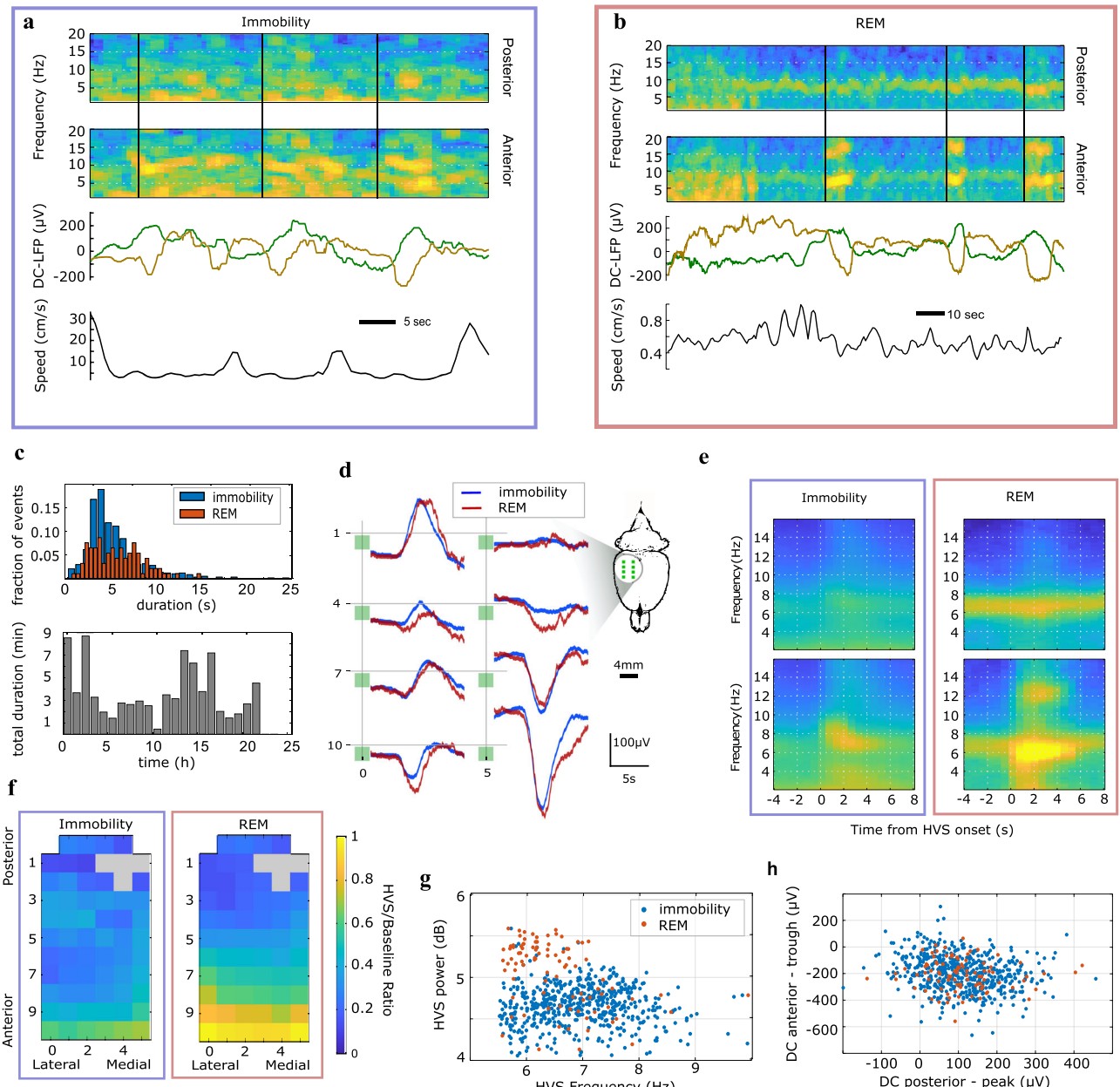

**Fig. 6 Topography of AC and DC dynamics associated with high-voltage spindles. a**, **b** examples of HVS events during immobility (**a**) and REM sleep (**b**). Spectrograms at posterior (top) and anterior (middle) positions on the array visualize dynamics of HVS (vertical lines at onsets) and ongoing oscillatory dynamics (low-frequency synchrony in IMM, theta oscillation in REM). Bottom two panels show DC LFP signals from posterior (green) and anterior (brown) positions on the array and head speed. **c** Distribution of duration of single HVS events for immobility and REM (top) and total hourly duration of HVS across recorded period (bottom). **d** Average DC LFP traces triggered on HVS onset during immobility and REM sleep (channels arranged as in Fig. 2g-left, inset showing anatomical localization of DC ECoG recording sites). **e** Average AC LFP spectrograms triggered on HVS onset during immobility (left) and REM sleep (right). The spectrograms at posterior (top) and anterior (bottom) positions on the array contrasting spectral content associated, respectively, with theta (posterior) and HVS (anterior) oscillations. Pseudo-color, spectral power (a.u.). **f** Topographic maps of change, relative to baseline, in HVS power at peak oscillation frequency during immobility (left) and REM sleep (right). **g** Distributions of power and peak frequency of individual HVS during immobility (blue) and REM sleep (red). **h** Distribution of DC fluctuation magnitude at posterior lateral and frontal medial derivations for single HVS during immobility and REM (color code as in **g**).

represents an important milestone towards the production of graphene-based neural probes at an industrial scale. On another front, the application of graphene transistors for neural sensing also requires the development of specific electronic equipment for biasing the sensors and converting the measured drain-to-source currents into equivalent voltage signals at the gate. In this study, we have presented a dedicated wireless headstage, which allowed describing the impact of the signal amplification and digitization

process on the sensitivity of the recording system. In this way, we have identified the challenges in the design of full-band amplifiers given specific energy and weight constrains for wireless applications. Furthermore, using a dedicated headstage we have demonstrated that the sensitivity of graphene active sensors in the ISA band is not significantly affected by the rest of the data-acquisition system components. Yet, we observed slow drifts in the measured signals in vivo, which were presumably attributed

to adsorption of charged molecules on the graphene channel or drifts in the electrode reference potential. These drifts were easily eliminated with a high pass filter at 1 mHz; however, future experiments could explore the functionalization of the graphene channel and use of alternative reference electrodes to maintain a more stable doping level.

Moreover, we have demonstrated the stability of the graphene sensors in vivo by characterizing their sensitivity over 4 weeks. In order to determine the stability of the signal quality we have also evaluated the signals induced by bipolar stimulation in the LFP frequency band, demonstrating a rather stable frequency response over time. These results suggest that the impedance of glial tissue surrounding the implant does not significantly affect the sensitivity of the g-SGFETs in vivo. To conclude the characterization of the device stability, we have evaluated the biocompatibility of graphene devices over 12 weeks via behavioral and histological markers. These results demonstrate an acute foreign body response comparable to platinum-based devices, which returns to values close to those of control animals 12 weeks after implantation. The demonstration of the graphene biocompatibility and long-term stability in a chronic implant represents another important turning point towards the large-scale production of graphene-based neural probes.

The experimental validation of this technology opens up many opportunities for electrophysiology studies in which having access to unconstrained behavior and multichannel recordings with sensitivity in a wide frequency band is relevant. In this study, we have shown quasi-continuous monitoring of brain activity in long recording sessions of up to 24 h, which allowed us to acquire large samples of neural activity across unperturbed behavioral and brain states. In combination with 3D-tracking, the wireless technology allowed us to explore the relation between neural activity patterns and behavior events, which occur sparsely over time, with sufficient statistical power. In particular, analyzing epicortical LFP signals with respect to rearing behavior we observed differential modulation of the 60–70 Hz gamma and 90–200 Hz high gamma range. While high-frequency activity between 90 and 200 Hz was consistently suppressed in a topographic manner, no such suppression was observed for the 60–70 Hz range, pointing towards distinct behavioral selectivity of underlying circuit mechanisms. The determination of frequency-specific power modulation in the gamma range, even for single rearing events, demonstrates the high sensitivity of the system in the high-frequency spectrum of the LFP.

In the low-frequency range, we found that infra-slow power <0.2 Hz increased significantly across DC sites during REM sleep episodes compared to SWS and thus showed the opposite state dependence than power in the slow oscillation band (1–4 Hz). Interestingly, infra-slow dynamics modulated power of theta and gamma rhythm during REM and with lower strength, power of sleep spindles during SWS. While modulation of LFP power in theta, beta and gamma bands by ISA phase derived from BOLD signal and DC EEG has been shown in humans[81,82] and recently in anesthetized rats[70], the present result is the first demonstration of interaction between physiologically established oscillatory dynamics, theta, spindle and gamma oscillations, and ECoG-derived ISA in freely moving rodent. Consistent with published intracranial work[73], volume conducted hippocampal theta measured at the cortical surface also modulated cortical gamma power, thus demonstrating that the developed technology is sufficiently sensitive to characterize known gamma dynamics.

While power of ISA modulating sleep spindle oscillations power during SWS was much lower than that during REM, a related hypersynchronous thalamo-cortical rhythm, HVS, was associated with much larger DC transients. Long-term wireless recording allowed us to evaluate the characteristics of HVS in a large statistical sample of events. The rate of occurrence during different brain states could be determined over a 24 h period, highlighting their occurrence also outside the awake resting state in which they are commonly reported to be selectively occurring. Specifically, we found them prominently expressed during REM sleep[83]. Having established this bimodal state specificity, we analyzed HVS events separately for awake immobility and REM sleep. Topographic analysis of the HVS peak power demonstrated that sensory-motor cortical preponderance for these oscillations is comparable for these two states. In contrast, the large sample size of events in both states allowed to identify a significant difference in both frequency content and power of the HVS between these two brain states. Furthermore, the capabilities to simultaneously map ISA patterns allowed to determine the distinct topographic structure of spatially specific infra-slow frequency components associated with HVS, showing phase-reversal across anterio-posterior axis. Importantly, these ISA features were conserved between REM-sleep and awake immobility, highlighting that the underlying origin of the DC signal is related to the HVS event itself and is independent of the brain state.

Future work is required to replicate the presented results in a large cohort of animals, follow them longitudinally and relate the surface pattern to intra-laminar and subcortical theta, spindle, gamma[77] and HVS[84] generators. Although we have focused the attention on epicortical ISA patterns, analyzing their correlation with ISA dynamics across cortical laminae could provide important insights into the origin and implications of ISA. A promising strategy in this direction is to combine the epicortical arrays presented here with graphene-based depth probes[33]. Future chronic recordings of depth and large-scale LFP signals across behaviors and behavioral states in freely moving unconstrained animals will lay the foundation for a new qualitative step in the brain dynamics investigation from infra-slow to very fast frequencies[72], contributing to our understanding of the origins of ISA dynamics in the context of resting states and default mode networks and its links to faster brain dynamics.

In summary, the thorough in vitro and in vivo evaluation of the sensing and long-term recording capabilities of graphene active sensors from a system perspective demonstrates the maturity of this technology and supports its application for the study of ISA without sacrificing high-frequency LFP components. In this direction, we have successfully evaluated ISA patterns during distinct brain states and the correlation of high-frequency oscillations with specific sparsely occurring behaviors. Our results represent an important step towards the broad implementation of graphene active-sensor arrays for neuroscience research, offering a stable and biocompatible sensing technology for long-term mapping of wide frequency band epicortical brain activity during spontaneous behavior.

## Methods

**Fabrication of g-SGFET arrays.** Arrays of g-SGFETs and devices for biocompatibility were fabricated on a 10 μm thick polyimide (PI-2611, HD Micro-Systems) film spin coated on a Si/SiO$_2$ 4" wafer and baked at 350 °C. Polyimide was chosen as a substrate due to its thermoxidative stability, high mechanical strength, insulating properties and chemical resistance[63,85], as well as its expected biocompatibility and previously reported stability for chronic implants[4,5]. A first metal layer (10 nm Ti/100 nm Au) was deposited by electron-beam vapor on a previously photodefined-negative AZ 5412E (Clariant, Germany) and then structured by a lift-off process. Afterwards, the graphene grown by chemical-vapor deposition on Cu was transferred (process done by Graphenea s.a.). In platinum devices for biocompatibility studies, another photolithography, metal evaporation and lift-off followed the first one. Graphene was then patterned by oxygen plasma (50 sccm, 300 W for 1 min) in a reactive ion etching (RIE). The photodefinable resist used to protect the graphene in the channel region was HIPR 6512, chosen to minimize the level of contamination. After the graphene etching, a second metal layer was patterned on the contacts following the same procedure as for the first layer. The lift-off step was followed by an annealing in ultra-high vacuum to improve the contact resistance and to eliminate resist residues from the graphene channel.

Subsequently, the transistors were insulated with a 3-μm-thick photodefinable SU-8 epoxy photoresist (SU-8 2005 Microchem), keeping uncovered the active area of the transistors channel. The SU-8 photoresist was chosen as insulating material because it is photodefinable and because its use in chronic implants has been previously reported[86,87]. The use of a photodefinable passivation polymer is required in the current graphene technology because etching of the passivation layer would also etch the underlying graphene channel. The polyimide substrate was structured in a reactive ion etching process using a thick AZ9260-positive photoresist (Clariant) layer as an etching mask. The neural probes were then peeled off from the wafer and placed in a zero insertion force connector to be interfaced with our custom electronic instrumentation. Finally, the devices were rinsed in ethanol to eliminate remaining resist residues on the graphene channel.

**Phase-amplitude coupling evaluation**. The signal inversion observed between channels in the infra-slow frequency band (0.005–0.05 Hz) was quantitatively evaluated by calculating the probability density of a signal amplitude as a function of its phase with respect to a second signal. In order to estimate the phase between the two signals the Hilbert transform of each of them was computed using the python library scipy and the difference between their phase calculated. A two-dimensional histogram was then used to express the probability density of the signal amplitude in the amplitude-phase space (Fig. 3c).

**Ethical approval and animal handling**. The experiments in-vivo were in accordance with the European Union guidelines on protection of vertebrates used for experimentation (Directive 2010/63/EU of the European Parliament and of the Council of 22 September 2010). Electrophysiological experiments with Long Evans rats were carried out under the German Law for Protection of Animals (TierSchG) and were approved by the local authorities (ROB-55.2-2532.Vet_02-16-170). Experimental procedures using Sprague Dawley rats for biocompatibility assessment were carried out according to the United Kingdom Animals (Scientific Procedures) Act, 1986 and approved by the Home Office and the local Animal Ethical Review Group, University of Manchester. Rats were kept under standard conditions (room temperature 22 ± 2 °C, 12:12 h light–dark cycle, lights on at 10:00), with food and water available ad libitum.

**Implantation of the graphene-sensor arrays for electrophysiological measurements**. As described previously in Garcia-Cortadella et. al.[34], an adult Long Evans rat, weighing 580 g, was anaesthetized with MMF (Midazolam 2 mg/kg, Medetomidin 0.15 mg/kg, Fentanyl 0.005 mg/kg). 1 h after, MMF induction Isoflurane was supplemented at 1% to maintain the rat anaesthetized and Metamizol was given at 110 mg/kg. The posterior-dorsal area of the head was shaved, the skin locally disinfected with Povidone-iodine and subcutaneously infiltrated with local anesthetic Bupivacaine. The skin was then incised and the dorsal skull cleaned carefully by blunt dissection. The dried skull was covered with UV-curing adhesive Optibond (Kerr) and a 3D-printed base ring was anchored to skull with screws and Metabond cement (Parkell).

Symmetric craniotomies with a maximum width of 5 mm were performed bilaterally, extending between +2 mm and −8 mm with respect to Bregma in the anterior-posterior axis. The dura mater was incised and removed within these craniotomies. A further craniotomy of 1 × 1 mm was performed over the cerebellum. All craniotomies were covered with prepolymerized polydimethylsiloxane (PDMS) (Sylgard 184, Dow Corning, USA) with mixing ratio 1:10 and sealed with Vetbond (Animal Care Products, USA). The skin margins around the implant were sutured and the implant closed with a protective cap.

After 1 week of recovery the g-SGFET array was implanted under Isofluran Anesthesia (5% induction 1% maintenance). After partial opening and sidewards flapping of the polymer covering the right hemisphere the array was placed onto the pial surface positioned such as to cover the posterior aspect of the right hemisphere (ca −7 to −2 mm from bregma). In addition, two Pt-Ir wires were implanted at either side of the g-SGFET array. One proximal to the array, the other distally on the opposing hemisphere. The polymer cover was flapped back into position with the flexible cable of the g-SGFET array leaving through the remaining slit. A second PDMS cover was used to cover both the incised polymer and array, anchored to the skull with Vetbond and Evoflow (Ivoclar Vivadent, Liechtenstein) and sealed with silicon gel 3-4680 (Dow Corning, USA). Finally, an Ag/AgCl electrode was placed in contact with the cerebellum as a reference for the recording of neural activity.

**Implantation of the graphene, platinum, and PI devices for biocompatibility evaluation**. Sprague Dawley rats (200–280 g) were anesthetized using isoflurane inhalation (typically at 3.5% for induction and between 1.5 and 2.5% for maintenance) in 100% oxygen. The top of the animal's head was shaved and the animal was positioned within a stereotactic frame with tooth and ear bar fixation. Animals were placed on a heated blanket, with a pulse oximeter attached to the foot and a rectal probe inserted to monitor body temperature. Viscotears liquid gel (Bausch & Lomb, UK) was applied to the eyes for protection during the procedure. Depth of anesthesia was confirmed and maintained throughout the surgery via absence of a pedal reflex. All experimental animals received a subcutaneous injection of buprenorphine (0.03 mg/kg). The head was swabbed with iodine and a large flap of

skin removed to expose the skull but not the temporal muscle. The periosteum was removed using a bone scraper. The skin around the perimeter of the removed tissue was glued to the bone using Vetbond tissue adhesive (3 M, UK). A craniotomy (~4 mm x 6 mm) was performed using a high speed surgical micro drill. Lambda was used as a posterior reference for the craniotomy, which was positioned at least 1 mm lateral to the midline, to avoid the sagittal sinus. The drilling region was rinsed regularly with saline to prevent heat damage. Once the bone around the border of the craniotomy was sufficiently thin, all bone shavings and other debris were removed using compressed air and the bone flap was gently removed. The cortical surface was kept moist using Ringer's solution. A fine needle with the tip bent at a 90° angle was used to gently lift the dura away from the cortical surface and another needle was used to create a slit in the dura, carefully positioned to avoid blood vessels. A pocket was created by lifting the dura next to the opening using fine forceps and the device was carefully placed on the cortical surface. The dura was then repositioned to hold the device in place. A glass window (UQG Optics, UK) of the appropriate size was positioned to fill the craniotomy and was fixed in place using dental cement (Superbond C&B, Prestige Dental). The animals received a subcutaneous injection of 0.9% saline (1 ml) and were placed in a recovery cage until the anesthetic had worn off.

**Behavioral testing for biocompatibility evaluation**. All animals had pre-surgical behavioral baselines taken at the age of 5 weeks. One week later, all animals were assigned to one of five groups; graphene electrode, platinum electrode, blank electrode, sham surgery (no electrode implanted), or naïve (no surgery). Animals were then tested at one or two timepoints—2 weeks, 2 and 6 weeks, or 6 and 12 weeks post-surgery. Timepoints were chosen in line with ISO 10993 definitions; where prolonged exposure is classified >24 h but <30 days, and permanent exposure is defined as >30 days (ISO 10993-6:2007).

Prior to the first exposure to NOR rats were placed into the empty arenas the day before testing for 20 min with their cage mates for acclimatization purposes. The square Plexiglas boxes (measuring 52 cm by 52 cm at the base with a height of 30 cm) had a white floor and black walls. Animals were acclimatized to the NOR arena before experiments began by placing them into the arena for 3 min while there were no objects within the arena. The NOR test comprises a training and testing trial, separated by an inter-trial period. In the training session, two identical objects were placed within the arena, such as two bottles of the same shape and size. In the testing session, two new objects were placed in the arena, one object identical to that during the training session, and one completely new object, such as a can. For training sessions, animals were placed in the arena and allowed to explore for 3 min, before returning to the home cage. Animals were left in the home cage for 30 min, before being placed into the arena for the testing session, again remaining in the arena to freely explore for 3 min. The time the animal spent interacting with the objects was measured in both the training and testing trials. In a healthy animal, the animal should spend more time interacting with the novel object during the testing session. Ideally these tests should not be used >3 times for any animal, and therefore animals in the 12 week implantation group were tested for NOR at baseline and then at 6 and 12 weeks post implantation. All other animals were tested at every selected timepoint available before sacrifice.

Videos of NOR test trials were manually scored by blinded researchers using an online stopwatch (http://jackrrivers.com/program/). Animals were classed as interacting with an object if their nose or paws touched the object. The amount of time spent interacting with the two objects was analyzed, and a discrimination ratio was determined by dividing time spent interacting with the novel object with time spent interacting with the known object. A discrimination ratio >0.5 indicated an animal had a preference for the novel object, a sign of normal cognition.

**Tissue collection and processing**. At 2 weeks, 6 weeks or 12 weeks post implantation of devices, animals were culled using an appropriate method for the type of subsequent analysis. For histology, animals underwent perfusion fixation using heparinised saline, followed by 4% paraformaldehyde (PFA; Sigma-Aldrich, UK; 441244) in phosphate-buffered saline (Sigma-Aldrich, UK; D8537). Tissue was stored a minimum of 24 h in PFA, transferred to a sucrose solution for 48 h, and frozen before cryosectioning 40 sections at 25 μm per animal. Cryosections were stained for one of three markers: (i) ionized calcium binding adaptor molecule 1 (Iba1) to quantify microglial population, (ii) terminal deoxynucleotidyl transferase dUTP nick-end labeling (TUNEL) staining to assess apoptosis, or (iii) haemotixylin and eosin (H&E) to assess gross morphology of the brain tissue. Tissue sections were blocked in 5% goat serum in PBS with 0.1% triton-X, before incubation with Iba1 primary antibody (1:200, 019-19741, Wako) overnight. A goat anti-rabbit Alexa 594 secondary antibody (1:1000, A11012, Invitrogen) was used for visualization. For immunofluorescently stained tissue, a DAPI counterstain was performed before slides were mounted using ProLong Gold mountant (P10144, ThermoFisher).

For TUNEL staining, the manufacturer's instructions were followed using a DeadEnd™ Colorimetric TUNEL System (G7360, Promega). Following diaminobenzidine (DAB) visualization of the TUNEL staining, slides were counterstained using methyl green (0.1% w/v aqueous solution, Alfa Aesar). H&E staining was performed as standard, using a 1-min haemotoxylin staining time, followed by an acetic acid rinse, and a 30-s eosin staining time. Slides for both TUNEL and H&E staining were mounted using DPX mountant (06522, Sigma-Aldrich).

Slides were imaged using the 3D Histec Pannoramic250 slide scanner, and images analyzed using CaseViewer (Version 2.2, 3DHistech Ltd). TUNEL-positive cells were counted and averaged across the cortical surface in forty 25 μm sections per hemisphere. Microglial cells were individually classified into one of four morphologies; Grade 0 (resting/ramified), Grade 1 (de-ramifying/re-ramifying), Grade 2 (activated/ameboid) or Grade 3 (clustered & activated) as previously described[88]. Activation was determined as a percentage of total microglial cells, which were either Grade 3 or 4.

For ELISA, animals were culled by rising concentration of $CO_2$, before cardiac puncture was performed to extract blood. Brain tissue was extracted, snap frozen in liquid nitrogen, and stored at −80 °C until further use. The extracted blood was collected in a blood collection tube (Vacutainer, Becton Dickson, UK) and allowed to clot at room temperature for 15–30 min. The tube was centrifuged at 5000RPM for 10 min at 4 °C, and the resulting serum supernatant was collected and stored at −80 °C until further use. There were insufficient serum samples to be run from the naive control group, so this group was excluded from analysis. Brain tissue was lysed by addition of liquid nitrogen and grinding the tissue to create a powder, to which NP-40 lysis buffer (150 mM NaCl, 50 mM Tric-Cl, 1% Nonidet P40 substitute, Fluka, pH adjusted to 7.4) containing protease and phosphatase inhibitor (Halt™ Protease and Phosphatase Inhibitor Cocktail, ThermoFisher Scientific) was added to the tissue. Samples were centrifuged at 5000RPM for 10 min, and the supernatant stored at −80 °C until further use. ELISA kits for four cytokines were used, IL-17a (437904, Biolegend), IFN-γ (439007, Biolegend), TNF-α (438204, Biolegend), and IL-6 (437107, Biolegend). Manufacturer instructions were followed for all four kits.

**Motion capture and behavioral states classification**. The rat was recorded for up to 24 h during spontaneous behavior in a recording arena of 100 × 100 cm to which it was prehabituated. During the recording session it had ad libitum access to food and water. The battery of the wireless system was exchanged once every 6 h. A motion capture (Mocap) system (Optitrack) using passive reflective markers and 8 cameras was used to track the motion of the animal head in three-dimensional space. Four reflective markers were anchored to the protective cap and their position averaged to infer the position in 3D and orientation of the head. Instantaneous headspeed in 3D was computed as time derivative of the modulus of the spatial coordinates. Motive 2.2 software was used for the analysis of Mocap data.

For the classification of the behavior as active or inactive, periods where headspeed exceeded 100 mm/s were labeled as active. In a second step active periods shorter than 5 seconds were skipped while gaps in active periods shorter than 5 s were concatenated with the neighboring active state. Timepoints, that did not fall under this definition of active, were labeled as inactive. The ratio of active and inactive periods varied substantially across total recording period (between 6.6% and 88.0% active per hour, mean 40.2% ± 4.7% active per hour).

Rear events were defined as short elevations of the head to heights that necessitate the animal to stand on its hindlimbs. After visual inspection of z elevations throughout the recordings a threshold of 200 mm elevation from ground was determined to effectively separate punctuate rearing onsets/offsets from ongoing height variations at lower z positions (see supplementary information S10). While active and inactive states were defined as mutually exclusive, rear events were considered to be a substrate of the active motor state.

**Neural signals processing and analysis**. The wireless headstage was controlled using Multi Channel Experimenter 2.12.1 software and the recorded data was converted to HDF5 format using Multi Channel Data Manager 1.13.1. Neural data was calibrated according to the transconductance of g-SGFETs using Python 2.7 scripts and exported to Neuroscope software for data exploration (see data and code availability statement). Analysis of neural signals was carried out using Matlab 2016b scripts (see code availability statement).

**Brain states classification**. Two channels of the epicortical array were chosen for the separation of brain states. One channel was selected from the posterior area of the array, putatively overlying the hippocampal formation and exhibiting prominent theta oscillations. A second channel was selected from the frontal area, recording from the region and neighborhood of somatosensory cortex, where high-voltage spindles are expressed most prominently. Power spectra were computed on whitened LFP signals in the range from 1 to 200 Hz using multitaper methods using 4-s windows in sliding 0.5 s steps. First, slow wave (SW) states were identified as periods were the z-score of the summed power of delta (1–4 Hz), alpha-beta ranges (10–25 Hz) exceeded −0.1. Gaps shorter than 5 s were concatenated with neighboring periods.

SW states coinciding with inactive motor state and >10 s temporal distance from the last preceding high-voltage spindle (HVS) event were defined as slow wave sleep (SWS). Following previous literature we assumed incompatibility between HVS and deep slow wave sleep in unanaesthetized Long Evans rats[84]. HVS are close in frequency to theta oscillations, but differ markedly in their expression of multiple higher harmonics due to their strongly non-sinusoidal waveshape. Owing to this fact spectral power in the frequency band from 20 to 50 Hz on the HVS reference channel was used to selectively detect HVS episodes and distinguish them from theta activity. The mean of multitaper power spectra from 20 to 50 Hz

was calculated and subsequently z-scored. Periods with z-score values exceeding 0.7 were labeled HVS. Candidate HVS periods shorter than 1 s were skipped to minimize false positives from occasional sharp single wave transients of undetermined physiological nature. Spectral profiles of the individual HVS events were post-hoc classified to remove artifact contamination. Onset and offset of peak power, instantaneous frequency and power of the first spectral peak were extracted from each event.

Theta states were defined based on the ratio between power in the theta (5–9.5 Hz) and delta (2–4 Hz) ranges on the theta reference channel. Gaps shorter than 10 s were concatenated with neighboring Theta periods. Theta coinciding with an inactive motor state was labeled Inactive Theta. All Inactive Theta periods that were preceded by a SW state within 1 s and which were longer than 5 seconds were considered REM sleep. All remaining Theta periods were considered Awake Theta. Finally, all periods that were neither SWS nor REM nor Awake Theta were defined as Awake Nontheta. It should be noted that the existence of low amplitude microstates during NREM sleep has been described previously[89]. It remains to be determined to which degree a subset of periods with low amplitude in the SW range assigned to Awake-Nontheta in this study maps onto states defined as low activity sleep microstates by Miyawaki and colleagues[89].

All states were considered to be mutually exclusive with the exception of HVS, which was considered an event occurring during but not interrupting ongoing background states. Therefore, for each HVS episode the brain states immediately preceding and following were merged if they belonged to the same state. All spectral analysis was performed using custom-developed Matlab implementation of multitaper estimate[90]. Analysis of the g-SGFET performance was done in Python. ISA brain dynamics analysis was limited to immobility and REM sleep, for which potential influence of motion artifacts on the infra-slow fluctuations could be discarded.

**Statistical analysis**. For the evaluation of the yield and homogeneity of the g-SGFETs performance, 9 neural probes with 64 g-SGFETs each were characterized in vitro. This data is plotted in the boxplots in Fig. 2, the boxes extend from the lower to the upper quartiles, with a line at the median. The whiskers extend 1.5 times the inter-quartile range and the data points beyond the whiskers are indicated by a dot. The longitudinal evaluation of g-SGFETs stability in-vivo was performed with one 64-channel array implanted on the cortex of a rat. The boxplots shown in Fig. 3 are defined as those in Fig. 2. In panels a and e of Fig. 3, the statistical sample are all 8 g-SGFET connected to the DC-coupled channels of the headstage. Panels d and f of Fig. 3 correspond to all 64 g-SGFETs on the array. Finally, panel g shows the normalized response of the 10 g-SGFETs on the positions under the highest induced electric field during bipolar stimulation.

For biocompatibility assessment, three device types; platinum, graphene, and polyimide (blank) were fabricated. The rats in each of these groups were implanted with one of the three device types on the parietal cortex of the brain. A fourth group of animals had the full surgery without the implantation of any device (sham control) and a fifth group (naive) had no surgery. For NOR testing, the number of rats used was $n = 7$ for all groups at all timepoints, except 12 weeks, which had $n = 3–7$ depending on the group. For cytokine detection in the brain tissue, the number of rats was $n = 4$ at 2 and 6 weeks, and $n = 3$ at 12 weeks post implantation. For microglial activation, the number of rats was $n = 3$ at 2 and 12 weeks and $n = 2$ at 6 weeks (or 3 for the contralateral hemisphere). In all cases, the contralateral hemisphere was also evaluated as a control. Data where $n = 3$ or higher were analyzed using a two-way ANOVA to compare all timepoints and interventions. Dunnett's multiple comparisons were then performed at each timepoint comparing each surgical intervention to the control. $*p < 0.05$, $**p < 0.005$, $***p < 0.001$, $****p < 0.0001$ are indicated for each surgical intervention in Fig. 4.

The measurement of gamma activity modulation during rearing was evaluated for 163 rearing events measured in the course of a 24 h recording in one rat. HVS were also measured during the same 24 h recording. Five-hundred sixty-six events were detected during immobility and 92 events during REM sleep. Differences in spectral content of HVS during distinct states (IMM and REM) was evaluated by a Wilcoxon ranksum test (the p-values are indicated in the main text). The electrophysiological data shown corresponds to one rat. The large statistical sample of events allows illustrating the capabilities of the graphene-based technology, however, the interpretation of these results should be subject to variability across measurements and animals.

The modulation of ISA power by REM state vs. SWS was evaluated in two ways: first, for the period directly around the state transition (−30 to 30 s around the REM onset). Second, we evaluated the ISA power in both states over their entire duration, not only in the SWS-REM transition. ISA power comparison between REM vs. SWS states was restricted to the 44 REM episodes lasting more than 40 s (see Supplementary Information S12B). To test for frequency-specific changes across the state transition we compared the distributions of median spectral power across trials for the 30 s pre vs. post SWS-REM transition for each frequency bin. Significance was assessed by permutation test for each frequency bin ($n = 1000$ permutations, see Fig. 5d example channel and Supplementary Information S12C for all working DC channels). Increase of ISA and concurrent decrease of 1–4 Hz power during SWS-REM transition is significant after permutation test ($n = 1000$ permutations) on all DC channels except one excluded channel, same as for longitudinal evaluation in Fig. 3a, e, due to poor signal to noise (Supplementary Information S12C). Second, we tested for statistical differences between

distributions of the integrated power in the ISA (0.01–0.1 Hz) band in SWS and REM states using Wilcoxon rank sum test (see Fig. 5e, test results for each channel in Supplementary Information S12D).

Modulation of the LFP power by the DC signal-derived ISA and LFP phase was quantified using instantaneous fast frequency power-weighted resultant length of the instantaneous slow frequency phase vectors normalized by mean LFP power in the respective band[72] for which magnitude reflects the strength of LFP power modulation to a preferred phase of the ISA or LFP. ISA and LFP phase and LFP power were computed as angle and absolute value of the analytical signal of the respective AC and DC channel signals that were band-pass filtered, with 0.04 and 0.4 Hz bandwidth, respectively. Significance of the modulation was tested based on 1000 surrogate phase-power pairs randomly shifted with respect to each other by up to 100 s. Resulting empirical p-value was corrected following false-discovery-rate control procedure at the error rate of 0.001. For constructing topographic maps of theta and spindle power band modulation we used mean modulation strength for the LFP power band 8–9 Hz (theta) and 10–14 Hz (spindle band), and the ISA phase frequency of 0.05-0.1 Hz computed for every AC channel (LFP power) and one fronto-medial DC channel (ISA phase).

**Reporting summary**. Further information on research design is available in the Nature Research Reporting Summary linked to this article.

## Data availability

Device characterization and raw electrophysiological data examples are available in GIN repository with the identifier (https://doi.org/10.12751/g-node.4zw2lt). The complete electrophysiological dataset is available from the corresponding author upon reasonable request. Source data are provided with this paper.

## Code availability

Custom code for the analysis of devices characterization is provided in GIN repository with the identifier (https://doi.org/10.12751/g-node.4zw2lt). The complete code for analysis of electrophysiological data is available from the corresponding author upon request.

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

## Acknowledgements

This work has been funded by the European Union's Horizon 2020 research and innovation program under Grant Agreement No. 732032 (BrainCom) and Graphene Flagship Grant Agreements No. 785219 (GrapheneCore2) and 881603 (GrapheneCore3). The ICN2 is supported by the Severo Ochoa Centres of Excellence program, funded by the Spanish Research Agency (AEI, grant no. SEV-2017-0706), and by the CERCA Program/Generalitat de Catalunya. R.G.C. is supported by the International Ph.D Program La Caixa-Severo Ochoa (Programa Internacional de Becas "la Caixa"-Severo Ochoa). This work has made use of the Spanish ICTS Network MICRONANOFABS partially supported by MICINN and the ICTS "NANBIOSIS", more specifically by the Micro-NanoTechnology Unit of the CIBER in Bioengineering, Biomaterials, and Nanomedicine (CIBER-BBN) at the IMB-CNM. This work is within the project FIS2017-85787-R funded by the "Ministerio de Ciencia, Innovación y Universidades" of Spain, the "Agencia Estatal de Investigación (AEI)", and the "Fondo Europeo de Desarrollo Regional (FEDER/UE)". A.S. and G.S. were also supported by Bundesministerium für Bildung und Forschung [grant number 01GQ0440]. R.G.C. acknowledges that this work has been done in the framework of the Ph.D in Electrical and Telecommunication Engineering at the Universitat Autònoma de Barcelona. We thank Eduardo Blanco Hernández for assistance with the preprocessing of the motion tracking data.

## Author contributions

R.G.C. fabricated the graphene sensor arrays and performed their characterization in vitro, he characterized the headstage and contributed to the recording and analysis of electrophysiological activity. G.S. developed the implantation methodology and did the surgery. He also contributed to electrophysiological signals recording and analysis. C.J. designed the wireless headstage and software for the operation of active sensor arrays. X. I. contributed to the design and fabrication of the neural probes. A.G., S.S., I.S., and E.S. performed the experiments for the biocompatibility study. E.M. fabricated and tested the devices for the biocompatibility study. K.K., S.S., and A.G., designed and interpreted the biocompatibility data. K.K. and S.S. overall coordinated the team for the biocompatibility study. A.G.B. contributed to the design of the recording system. A.S. coordinated the experiments for electrophysiological data-acquisition and contributed to the analysis of electrophysiological signals. J.A.G. coordinated the team in the ICN2 for the fabrication of the sensor arrays. R.G.C., G.S., S.S., A.S., and J.A.G. wrote the manuscript.

## Competing interests

Patent application (no. P201831068) filled by CSIC, ICREA, CIBER, ICN2, and IDI-BAPS; concerning a graphene transistor system for measuring electrophysiological signals (pending); inventors who are co-authors in the present article are A.G.B., X.I., E.M., and J.A.G.
