## [Peer Review File · Nature Communications]

REVIEWER COMMENTS

Reviewer #1 (Remarks to the Author):

In this manuscript, Garcia-Cortadella et al. aimed to characterize the compatibility of graphene-based transistors as transducers of neural activity in rodents using previously established rodent electrophysiological experiments and equipment. The authors' main motivation arose from their interest in graphene and graphene-based devices, with the goal of expanding application of graphene into neuroscience.

The authors' reasoning as to why a transistor interface such as their g-SGFET is good candidate for the building block of a large-scale neural interface device is that they can record so called infra-slow activities (ISA). Previously, they demonstrated the capacity of such a transistor to record ISA (Masvidal-Codina et al Nat Mat, 2018), and used it in a multiplexed acquisition system (Schaefer et al 2020 2D Mat).

In this manuscript, they primarily focus on reproducibility, consistency and stability of their array. Although it is an incremental step, and there is no new neuroscience finding enabled by g-SGFET, it is certainly important to clearly and carefully evaluate the capacity of graphene-based electrolyte transistors in neural interface devices. Please see below for my concerns and suggestions regarding the manuscript. I hope my comments will help the authors to better present their work and result into an improved manuscript.

What is the advantage of graphene-based electrolyte gate FET (EGFET) compared to other transistors that have already been used in vivo, and in particular for neural signal transducers? There are numerous examples of such devices made out of organic and inorganic materials, as well as in the form of field-effect or electrochemical architectures. In its current state, the introductory section of the manuscript describes this work only in the context of graphene, with no connection to other efforts which helped established the field. This makes the manuscript rather dismissive of other work, resulting in misleading statements and conclusions, despite the strong expertise and history of authors in the fields of material science and neuroscience. I think it would be important to know what is the current state of the field in regards to transistor technology in such a manuscript where the main focus is transistor applications.

How does the performance of g-SFET compare to other transistors used in vivo, such as organic and silicon transistors? The transconductance here is about 1mS and ON/OFF ratio of about 3.5. Usually transistors should have at least several orders of magnitude ON/OFF ratio. Considering the fact that such transistors are used as the transducer/buffer such low ON/OFF may significantly lower the dynamic range of the signal. What are the strategies to improve the device performance?

The probe itself consists of PI substrate (10 μ m thick) and a photoresist (SU-8 3 μ m thick) as insulator. What was the reasoning in using such materials as they are known for their lack of longer term stability and tendency to absorb water over time? I assume the authors mitigated this issue by using a thicker substrate layer which in turn makes the device less conformable. In the current setting, the transistors and interconnects are not in the mechanical neutral plan. What are the microfabrication challenges associated with utilizing thinner substrates and softer material with graphene?

The authors use the term "wide-band" as a capacity of their transducer. Usually, such term in electrophysiology refers to recording from both local field potential and action-potential (AP) of neural activity for the purpose of understanding their relationship. For example, this working definition is used in the work of Sirota et al Neuron 2008 (REF 53, one of the co-authors). Therefore, it is critical for the authors to demonstrate the capacity of their device in simultaneously recording LFP and APs together, in vivo. Furthermore, epileptic activity often takes the form of high frequency oscillations (~200-400Hz). What is the bandwidth of the transistor array used here? If it is adequate for recording

spikes and high frequency oscillations, could the authors elaborate as to why they limit the system to investigating gamma oscillations?

" g_m is proportional to the graphene-electrolyte interface capacitance per unit area, but not to the impedance of the active area" Hence, it can record slow frequencies better compared to a microelectrode that has a very high impedance at lower frequencies. I have a few questions:

1- Can all electrolyte-gated transistors record the same phenomena? Why is a graphene transistor more suitable for this task?

2- Geometrical variation of the transistor channel will affect both G_m and Z . In fact, drain current in a field effect transistor, hence G_m , is a function of gate capacitance (Z of channel/electrolyte). This statement requires experimental evidence. Furthermore, one can argue that g-SGFET can be described as an electrode (equal size of to the channel) + FET amplifier as long as one can monitor drain current at low frequencies.

The authors nicely integrated their transistor with existing tools and acquisition systems used in neuroscience, and such efforts are instrumental for technology dissemination. As a result of this integration authors were able to establish acquisition of ISA in their previous research in which is the major focus of the introduction and importance of g-SGFET. However, limited neurophysiological analysis was performed here around ISA itself and its relevance to other electrophysiological activity or functions. The authors mostly focused on higher frequencies such as gamma oscillations and high voltage spindles that can be observed and analyzed using traditional multi-electrode arrays. It is certainly encouraging to be able to observe similar activity to what current electrodes can provide but considering the strong emphasis of the technology and its advantages in ultra-low frequency regime I think it would be important to investigate ISA events. I understand that discovery science is not the focus of the manuscript and it is not reasonable to request a discovery while the authors provided extensive yield, biocompatibility and integration results. However, perhaps simple analysis such as demonstration of ISA fluctuation over sleep states or changes of ISA power during physiological events not previously characterized in the infra-slow regime would dramatically strengthen the manuscript, making their technology more attractive and applicable.

The authors' effort in providing detailed statistical analysis, population data and reproducibility of the data is certainly appreciated. Such studies have the potential to improve our understanding of brain function and create unique interdisciplinary knowledge.

Reviewer #2 (Remarks to the Author):

The authors report on the use of graphene transistors that demonstrate long-term and wireless mapping of wide-bandwidth brain activity which together with a detailed noise analysis are the main findings of the manuscript. The manuscript is very interesting and demonstrates the possible applications of transistors in neurophysiology beside bioelectronic applications of OECTs. The authors, who characterized the noise behavior of graphene transistors in great detail, have clearly worked out the advantages of these electronic components over conventional microelectrode systems in electrophysiological recordings.

Minor points:

Results

Homogeneity and sensitivity of graphene active sensor technology:

- which frequency bandwidth was used for V_{gs-rms} measurements
- the authors have excluded outliers in the gate equivalent noise representation from their analysis. Is this correct? What is the reason?
- it is difficult to verify the low rms noise of 4 μV in Fig 2d taken the wide distribution of in Fig. 2b and I would expect rather an average of 10 -15 μV for V_{gs-rms}
- Fig 2 h is misleading: it looks like the time domain signals measured in the DC mode are much

smaller than those measured in the AC mode. I expect rather an opposite behavior from the 1/f noise

Signal stability and sensitivity over time:

- Fig. 3a shows the shift in the CNP over a time period of 4 weeks. Was a shift towards more neg V_{GS} also observed or only a shift towards more pos values? How much is the shift of the reference electrodes during this time period?

Reviewer #3 (Remarks to the Author):

The authors have developed and tested a 64-ch flexible electrode array incorporating flexible graphene transistors. Graphene is a promising material for future neural interface technologies, particularly for its use in multiplexed recording systems that enable the number of electrodes to scale without increasing the number of external wiring connections. In this work, the authors present the yield and performance uniformity of graphene active sensor arrays fabricated in their cleanroom. They also present chronic, in vivo, wireless recording for 30 days after implant and histology confirming preliminary biocompatibility of the arrays after 12 weeks. The work is important and well done. Congratulations on a very good paper.

I have a few questions and small edits for the authors to address.

Questions:

1. Can you confirm the size of the electrode / graphene gate / neural sensing area? Is it 100 μm x 100 μm ? That wasn't clear to me.
2. Related, you mention that graphene active arrays are able to better record infraslow brain signals (<0.5 Hz), but I don't see any data to confirm this. In our experience, 200 μm diameter PtIr electrodes have low enough impedance and stability that makes them capable of recording ultra-slow signals, down to 0.005 Hz, with very low noise. Can you compare the in vitro noise levels (<0.5 Hz) of the graphene active devices to comparably sized passive electrodes? Perhaps using Pt or PtIr or Pt Black? Are there any other factors that make graphene active arrays more capable of recording ISA than traditional faradaic passive electrodes?
3. Line 183 - Are you using the full scale of the ADC with the gain provided? It seems like it would only use a small part of the ADC scale, given the red curve in figure 1b. A 1mV p-p signal should yield approximately 100 nA signal, yielding a 12 mV signal after IV conversion of 12,000 and a voltage gain of 10. Is that correct?
4. Supplementary materials S2 - S2. Intrinsic 1/f noise and headstage noise, line 68. I thought this should be 1/A not 1/sqrt(a) because the formula for flicker noise is:
$$V_2 / \text{Hz} = K / (\text{Cox} * W * L * f)$$

All lines refer to main text unless otherwise stated

Small edits

1. Line 29 - "maturity", there may be a better way to say this or define what you mean.
2. Line 64 - Should say "curves at a particular V_{ds}" rather than "curves over V_{ds}"
3. Line 180 - Should say fig 2E instead of 2D
4. Line 182 - Should say fig 2E instead of 2D
5. Line 185 - Should say fig 2E instead of 2D
6. Line 202 - Should say Fig 2H instead of 3H
7. Figure 1f - would be good to include spacing and size of electrode sites
8. Line 338 - why did you choose these particular cytokines?

9. Line 450 - Should this be LFP instead of LPF?
10. Line 453 - small error but this should be Figure 5

Dear Editor, dear reviewers,

Thank you for your constructive review, which has helped to significantly improve our manuscript. We have addressed all questions and implemented the required changes, as detailed in the following.

Reviewer #1 (Remarks to the Author):

In this manuscript, Garcia-Cortadella et al. aimed to characterize the compatibility of graphene-based transistors as transducers of neural activity in rodents using previously established rodent electrophysiological experiments and equipment. The authors' main motivation arose from their interest in graphene and graphene-based devices, with the goal of expanding application of graphene into neuroscience.

The authors' reasoning as to why a transistor interface such as their g-SGFET is good candidate for the building block of a large-scale neural interface device is that they can record so called infra-slow activities (ISA). Previously, they demonstrated the capacity of such a transistor to record ISA (Masvidal-Codina et al Nat Mat, 2018), and used it in a multiplexed acquisition system (Schaefer et al 2020 2D Mat).

In this manuscript, they primarily focus on reproducibility, consistency and stability of their array. Although it is an incremental step, and there is no new neuroscience finding enabled by g-SGFET, it is certainly important to clearly and carefully evaluate the capacity of graphene-based electrolyte transistors in neural interface devices. Please see below for my concerns and suggestions regarding the manuscript. I hope my comments will help the authors to better present their work and result into an improved manuscript.

We thank the reviewer for a very constructive review, which has helped us to improve the manuscript. We agree with the reviewer's general statement about the main focus and objective of our work. Certainly, the focus of this work is on the advancement and validation of a graphene-based brain mapping technology in chronic long-term recordings, revealing possible applications via example cases, setting the stage for but not focusing on neuroscience discoveries.

We would like to point out that our technology-oriented study on the sensitivity, reproducibility, stability and biocompatibility of g-SGFET arrays represents a necessary step towards the application of these devices in chronic electrophysiology studies in freely-moving animals. The presented results highlight the maturity of both the g-SGFET technology and the acquisition electronics, which have to be properly adapted for acquiring signals from active transducers.

Although the focus of the article is indeed not on any particular neuroscience discovery, we would like to emphasize the novelty of the multiple results related to infra-slow signals, such as ISA power change between sleep states, ISA phase modulation of theta, spindle and gamma oscillations, mapping of infra-slow signals related to high-voltage spindles, which are allowed by the capabilities of this technology to monitor infra-slow signals. We believe that the proof-of-concept study that we report in this work calls for future experiments and illustrates the potential of this technology to the neuroscience community.

What is the advantage of graphene-based electrolyte gate FET (EGFET) compared to other transistors that have already been used in vivo, and in particular for neural signal transducers? There are numerous examples of such devices made out of organic and inorganic materials, as well as in the form of field-effect or electrochemical architectures. In its current state, the introductory section of the manuscript describes this work only in the context of graphene, with no connection to other efforts which helped established the field. This makes the manuscript rather dismissive of other work, resulting in misleading statements and conclusions, despite the strong expertise and history of authors in the fields of material science and neuroscience. I think it would be important to know what is the current state of the field in regards to transistor technology in such a manuscript where the main focus is transistor applications.

By no means our intention has been to be dismissive of earlier works in the field. Some of us have been involved in the exploration of other transistor technologies for bioelectronics, including more standard semiconducting materials. Previous works from our teams have provided a general comparison graphene-based sensors with other material systems¹⁻⁴, extensively citing prior studies. That is one of the reasons why our review of previous literature in this manuscript was initially not so extensive. Yet, we agree with the reviewer that the contextualization of our work as part of the current state of active sensors technologies for neuroscience can be improved. Thus, we have slightly expanded the technology overview presented in the first paragraph of our manuscript in order to cite recent advances in other material systems as well as referring to recent specialized review articles.

“Increasing the bandwidth of neuroelectronic interfaces in terms of spatial resolution and sensitivity in a wide frequency range is a major and ongoing challenge in neural engineering. In the last decades, large efforts have been dedicated to the development of neural sensing interfaces with high sensor-count on conformal substrates⁴⁻¹³, which are required for highly biocompatible intra-cranial neural probes¹⁴⁻¹⁷. In this line, ~~graphene~~ active sensors have emerged as a promising building block for high bandwidth neural interfaces^{2,4,8,18-21} because they can be arranged in a multiplexed array^{4,6,8,10-12} enabling high sensor-count probes. ~~The detection principle of active sensors is typically based on the modulation of the conductivity of a transistor channel, which is electrically coupled with the biological environment through its gate^{2,11,12,20,22-24}, producing a local signal pre-amplification. Although active sensing technologies present substantial advantages over conventional micro-electrode arrays, their implementation is currently limited by the demanding material properties required. In order to achieve long-term and highly sensitive neural recordings, materials for active sensing neuroelectronic interfaces are expected to exhibit semiconducting or semimetallic properties, a high electrical mobility and low intrinsic noise, in addition to a high stability, easy integration in flexible substrates and biocompatibility. Some active sensors based on organic semiconductors and thin Si nanomembranes have exhibited promising performance, with novel transistor architectures^{19,24} and insulating technologies^{8,17} improving their performance in some typically constrained aspects such as their frequency response or their long-term stability. Graphene-based active sensors are another promising candidate for neural interfaces due to the flexibility of graphene^{25,26}, its high expected stability²⁷ and biocompatibility^{28,29} as well as its outstanding electronic properties, including an extremely high mobility of charge carriers^{30,31}. Graphene solution-gated field-effect transistors (g-SGFETs) have demonstrated a high sensitivity for the detection of local field potentials² (LFP) as well as a high performance in multiplexed operation^{4,10}. In addition, g-SGFETs have recently demonstrated an outstanding sensitivity for the mapping of infraslow (<0.5Hz) brain activity (ISA)³²⁻³⁴ with high spatial resolution^{3,4,10,35}.”~~

How does the performance of g-SFET compare to other transistors used in vivo, such as organic and silicon transistors? The transconductance here is about 1mS and ON/OFF ratio of about 3.5. Usually transistors should have at least several orders of magnitude ON/OFF ratio. Considering the fact that such transistors are used as the transducer/buffer such low ON/OFF may significantly lower the dynamic range of the signal. What are the strategies to improve the device performance?

Comparing the g-SGFET with silicon and organic transistors, a remarkable difference is indeed the ON/OFF ratio, which presents very low values in the g-SGFET due to the semimetallic properties of graphene, with zero bandgap and a vanishing density of states at the Dirac point. This lack of a suitable ON/OFF ratio is the main reason why there is a general acceptance in the field of electronics that graphene is not suitable for digital electronics, as we know them now.

Nevertheless, graphene presents an extremely high electrical mobility, which makes it suitable for analog electronics, including active sensors. In particular, it is possible to design solution-gated field effect transistors in which the graphene channel is in direct contact with an electrolyte solution. This concept, and the advantages and disadvantages with respect to other material systems has been addressed in previous publications, see for instance Hess et. al. IEEE (2013) and Hébert et. al. Adv. Funct. Mater. (2017). The neural signal transduction by the g-SGFETs is described by their transfer characteristics (see Fig. 1b in the main text and Fig. 1 below).

Regarding the dynamic range of the transducer, the limiting factor is the voltage window in which the transfer characteristics present a linear response, rather than their ON/OFF ratio. In fact, we

have recently published a detailed evaluation of the effect of non-linearities in the transfer characteristics of g-SGFETs on the harmonic distortion of the transduced neural signals³⁵ (see Fig. 1a below). Our results demonstrate a negligible effect for signal amplitudes below few mV in a wide gate-bias range, as represented by the distortion-to-noise ratio in Fig. 1b below. Typically, extracellularly recorded neural signals are in the low mV range.

Figure 1: a. Typical transfer characteristics of g-SGFETs. The harmonic distortion of large amplitude signals is illustrated. b. Distortion-to-noise ratio (DNR) with the noise integrated in a narrow (4-40Hz band) represented in the signal amplitude-gate bias ($A_{sig} - V_{gs}$) space. Contour lines represented by the solid lines every 20dB. Adapted from Garcia-Cortadella et. al. Small (2020).

Regarding the transconductance, in the case of the graphene solution-gated field-effect transistors, typical values are 1-4 mS/V (for transistors with $W=L$, see Fig. 2a in the main text), which results from the relatively high capacitive coupling of the graphene with the electrolyte (typical graphene-electrolyte capacitance² of 1-2 $\mu\text{F}/\text{cm}^2$) and from the high electrical mobility of carriers in CVD graphene ($\sim 1000\text{-}2000\text{ cm}^2/\text{Vs}$ in our devices). On the other hand, in the case of organic semiconductors, different device configurations have been used in the past, electrochemical transistor configuration (OECT) and electrolyte gated organic field effect transistors³⁶ (EGOFET). In the case of the EGOFETs, typical transconductances are very low because of the low electrical mobility in organic semiconductors. OECT devices, though, present volumetric capacitance and are known for presenting a very high double-layer capacitance. However, their mobility is typically much lower than in CVD-graphene. Therefore, typical transconductance values for organic electrochemical transistors vary in a great extent depending on the channel thickness, from values in the same range as for g-SGFETs to much higher values^{19,37}. On the other hand, flexible silicon FETs present transconductance values about 200 times lower than for g-SGFETs¹¹ with transconductance of $\sim 0.1\text{ mS}/\text{V}$ for $W=10L$.

Having a relatively high transconductance is important in order to pre-amplify the signals above the noise floor of the transimpedance amplifiers typically used to convert the current variations of the transistors into a voltage signal. However, active sensors also present an intrinsic, typically 1/f, noise which scales with the drain-to-source current. Therefore, the most important figure of merit which must be optimized to increase the sensitivity of active sensors is the G_m to RMS current noise (I_{ds-rms}) ratio, referred to as equivalent noise at the gate (V_{gs-rms}). Regarding the development of strategies to optimize the sensor performance, we have recently published several articles on the modelling and mitigation of intrinsic noise in graphene devices^{38,39}. In the present manuscript, we

report novel results on the V_{gs-rms} of g-SGFETs evaluated from a large statistical sample and from a system perspective.

To address this comment of the reviewer, we have modified the discussion about these aspects in the section “Homogeneity and sensitivity of graphene active sensor technology”:

“The measure median g_m , 1.9mS/V, is relatively high with respect to flexible silicon FETs¹¹ and comparable with typical organic transistor values^{19,37} due to the high electrical mobility and gate capacitance of g-SGFETs.”

and in the section “Wireless headstage design and characteristics”:

“Having a relatively high transconductance is important in order to pre-amplify the signals above the noise floor of the transimpedance amplifiers. However, active sensors typically present an intrinsic 1/f noise, which scales with the drain-to-source current³⁹. Therefore, V_{gs-rms} is a more suitable figure of merit to evaluate the sensitivity of active sensors. In order to validate that the sensitivity of the recording system is limited by the intrinsic noise of the active sensors, it is paramount to evaluate the impact of the amplification electronics on the sensitivity of the system in a wide frequency band.”

The probe itself consists of PI substrate (10 μ m thick) and a photoresist (SU-8 3 μ m thick) as insulator. What was the reasoning in using such materials as they are known for their lack of longer term stability and tendency to absorb water over time? I assume the authors mitigated this issue by using a thicker substrate layer which in turn makes the device less conformable. In the current setting, the transistors and interconnects are not in the mechanical neutral plan. What are the microfabrication challenges associated with utilizing thinner substrates and softer material with graphene?

Polyimide and SU-8, together with parylene-C, are the most commonly used polymers for neuroelectronic interfaces (in the field of neuroscience research, where typical chronic studies are well below 1 year) because of their compatibility with microfabrication techniques and low Young’s moduli¹⁸. Parylene-C is among them the one with the lowest moisture absorption¹⁸, however, polyimide has also been used successfully as a substrate and passivation layer for chronic implants in neuroscientific studies^{8,9}. As a disadvantage, both parylene-C and polyimide are not photodefinable. Although there are photodefinable formulations of polyimide, their water uptake is worse than for polyimide and should be avoided for chronic implantation⁴⁰. We agree that the use of parylene-C for the passivation would be desirable, however SU-8 was chosen because it is photodefinable and because its stability for chronic implants has also been previously reported^{41,42}. The reason a photodefinable passivation polymer is required in the current graphene technology is that etching of the passivation layer is not compatible with the underlying graphene channel. On the other hand, peeling off sacrificial parylene-C layers, which is typically used for OECTs fabrication, can leave metal contacts exposed to the tissue in the actively biased sensors, ultimately causing faradaic reactions at the metal-electrolyte interface. Therefore, the combination of polyimide and SU-8 represents a suitable platform to evaluate the minimal long-term stability of g-SGFETs.

However, we agree the intrinsic long-term stability of graphene might be beyond the reported values. Currently, efforts are being dedicated to develop a g-SGFET technology compatible with non-photodefinable polyimide by protecting the graphene channel against dry etching of the passivation layer⁴³. Similarly, we agree that the position of the graphene channel with respect to the neutral plane might affect negatively the stability of the active sensors. Yet, our experiments reveal a rather high stability, which sets a lower bound for the g-SGFETs stability.

We acknowledge the comment of the reviewer, which highlights the potential improvements in the technology by using alternative polymer materials for the substrate and passivation layers. For very long-term clinical applications, in which the devices must remain implanted for years, some groups have been dedicating a great effort in developing advanced passivation methods¹⁷, which could be eventually translated to the neuroelectronic devices under development.

We have highlighted that the reported results represent a lower bound for the g-SGFETs stability in the main text:

“Furthermore, the polymers used as a substrate and passivation layers could be modified to reduce the moisture absorption^{17,40} and displace the neutral plane of the device at the position of the graphene channel (see Methods section). Yet, the results presented in this section reveal a promisingly stable performance over time, which sets a lower bound for the stability of g-SGFETs in a chronic implant environment.”

and we have included a note to clarify the reasons for this choice of substrate and passivation polymers in the Methods section “Fabrication of g-SGFETs”.

“Arrays of g-SGFETs and devices for biocompatibility were fabricated on a 10 μm thick polyimide (PI-2611, HD Microsystems) film spin coated on a Si/SiO₂ 4” wafer and baked at 350°C. Polyimide was chosen as a substrate due to its thermoxidative stability, high mechanical strength, insulating properties and chemical resistance^{40,44}, as well as its expected biocompatibility and previously reported stability for chronic implants^{8,9}.”

“Subsequently, the transistors were insulated with a 3-μm-thick photodefinable SU-8 epoxy photoresist (SU-8 2005 Microchem), keeping uncovered the active area of the transistors channel. The SU-8 photoresist was chosen as insulating material because it is photodefinable and because its use in chronic implants has been previously reported^{41,42}. The use of a photodefinable passivation polymer is required in the current graphene technology because etching of the passivation layer would also etch the underlying graphene channel.”

The authors use the term “wide-band” as a capacity of their transducer. Usually, such term in electrophysiology refers to recording from both local field potential and action-potential (AP) of neural activity for the purpose of understanding their relationship. For example, this working definition is used in the work of Sirota et al Neuron 2008 (REF 53, one of the co-authors). Therefore, it is critical for the authors to demonstrate the capacity of their device in simultaneously recording LFP and APs together, in vivo. Furthermore, epileptic activity often takes the form of high frequency oscillations (~200-400Hz). What is the bandwidth of the transistor array used here? If it is adequate for recording spikes and high frequency oscillations, could the authors elaborate as to why they limit the system to investigating gamma oscillations?

We acknowledge the comment of the reviewer, as we recognize the importance of keeping a consistent terminology in the field. Although the term wide-band has been indeed used for electrophysiological recordings covering the LFP and AP bands, it has also been used to describe the simultaneous recording of infra-slow frequency components and higher-frequency LFP bands³, which is the meaning we implied by “wide-band” here. However, we agree that using the term wide-band might lead to confusion and therefore we have replaced it by the term “wide frequency-band”, which is merely descriptive and has fewer implications. Regarding the sensitivity of our system to detect APs; to our knowledge APs have only been detected from the brain surface using ultra-sensitive small area PEDOT:PSS electrodes⁵, and their detection is therefore not a standard in epicortical recording systems. Today, the above-cited publication is the only report of successful, but not routine, AP recordings. To avoid confusion, we have added the term “epicortical” before “brain/neural activity” both in the title of the manuscript as well as throughout the main text and have explicitly described the frequency range covered by the present epicortical sensing technology in the abstract.

“Furthermore, to illustrate the potential of the new technology to detect cortical signals from infra-slow to high-gamma frequency bands, we perform proof-of-concept long-term wireless recording in a freely behaving rodent.”

Regarding the detection of high frequency epileptic activity, we have validated the presented recording system in an animal model which is not prone to high-frequency epileptic seizures, at least at the cortical surface. Thus, recording of such pathologic activity was not possible in the reported experiments. Instead, we focused our analysis on high-gamma (90-200Hz, see Fig. 5i-j) as an

illustrative example of high-frequency physiologic signals correlated with sparse behavioural events, demonstrating the detection capabilities at the single-trial level. In the newly added results (Fig. 5h) we demonstrate that g-SGFET electrodes can detect fluctuations of different frequency gamma power generators modulated by infra-slow and theta rhythms. Latter result replicates prior intracranial recordings work and thus shows that our methodology is sufficiently sensitive.

Regarding the bandwidth of the system, we have presented a longitudinal characterization of the g-SGFETs bandwidth ($G_m(f)$) *in-vivo* for frequencies up to 2.15kHz and over 4 weeks (see Fig. 3g in the main text). Detection of APs *in-vitro* using g-SGFETs has been previously reported⁴⁵ and the fractional order G_m attenuation and its effect on the sensitivity of g-SGFETs at high-frequencies has been thoroughly discussed³⁵.

“ g_m is proportional to the graphene-electrolyte interface capacitance per unit area, but not to the impedance of the active area” Hence, it can record slow frequencies better compared to a microelectrode that has a very high impedance at lower frequencies. I have a few questions:
1- Can all electrolyte-gated transistors record the same phenomena? Why is a graphene transistor more suitable for this task?

For a detailed discussion we kindly refer the reviewer to the recent article by Masvidal et. al.³. Here we only include a short summary.

The first reason for an improved sensitivity with respect to passive electrodes is the use of the field-effect principle. Given a certain static electric field applied at the channel-electrolyte interface, a measurable stationary carrier accumulation is produced, which is proportional to the applied electric field. By using this detection mechanism, the voltage divider between an electrode impedance and the input impedance of the amplifier can be prevented. This advantage is expected to hold for any active sensor with **stable transfer characteristics**. A second reason for the improved sensitivity is the rather stable electrochemical potential at the graphene-electrolyte interface, due to the chemical inertness of graphene^{3,27}.

So far, no research has been reported on the sensitivity of other active sensors to detect infra-slow signals. In the case of organic semiconductors, evaluation of electric potential over time for passive electrodes *in-vitro* shows significant drifts⁴⁶, which might hamper their sensitivity in the ISA band. In our work, we present accurate values for the sensitivity of g-SGFETs in the infra-slow band, including an evaluation of the effect of potential drifts *in-vivo*, which represent a solid reference to evaluate the sensitivity of other active sensor technologies in the infra-slow frequency band.

We have included a comment in the introduction of our manuscript to clarify which is the potential of other active sensors to detect infra-slow activity.

“Signal detection based on the field-effect mechanism therefore allows to prevent the signal distortion and gain loss observed for small passive sensors in the infra-slow frequency band. This advantage is expected to be valid for all FET-based sensor technologies with stable transfer characteristics, however, experimental proof has been only shown for g-SGFETs, which present a particularly high chemical inertness^{3,27}.”

2- Geometrical variation of the transistor channel will affect both G_m and Z . In fact, drain current in a field effect transistor, hence G_m , is a function of gate capacitance (Z of channel/electrolyte). This statement requires experimental evidence. Furthermore, one can argue that g-SGFET can be described as an electrode (equal size of to the channel) + FET amplifier as long as one can monitor drain current at low frequencies.

Indeed, G_m depends on the gate capacitance per unit area (intensive property) and the W/L ratio of the transistors, but not on their active area. We have slightly modified this statement in the manuscript to clarify the meaning (see below).

In the aforementioned statement, we cited an article⁴⁷ which describes a current-voltage model of g-SGFET to further clarify the geometrical dependence of their transconductance. The following definition of G_m is common to all field-effect transistors:

$$G_m \equiv \left. \frac{dI_{ds}}{dV_{gs}} \right|_{V_{ds}} = V_{ds} \frac{W}{L} \mu e \frac{dn}{dV_{gs}}$$

Where e is the charge of an electron, W/L the width to length ratio, μ the electrical mobility and n the density of charge carriers per unit area (the rest of variables are defined in the main text). In graphene FETs, away from the Dirac point (in the linear range of the transfer characteristics);

$$\frac{dn}{dV_{gs}} = \frac{d}{dV_{gs}} \frac{C_{gate} V_{gs}}{e}$$

where C_{gate} is the gate capacitance per unit area (intensive property). Note that the only geometrical dependence of G_m is thus described by the factor W/L , with G_m being independent of the active area of the transistor.

Another argument, which supports our initial statement, is that G_m is ideally frequency independent. That is because it depends on the capacitance per unit area, but not on the impedance as an extensive property, which is frequency dependent.

Regarding the request of the reviewer for experimental evidence; experimental proof was provided in the work by Masvidal et. al.³ (supporting information). We acknowledge the comment of the reviewer and we have now clarified this statement in the main text.

“ g_m is proportional to the gate capacitance per unit area (intensive property) and to the W/L ratio of transistor, but not to its active area^{19,39,47-49}.”

The authors nicely integrated their transistor with existing tools and acquisition systems used in neuroscience, and such efforts are instrumental for technology dissemination. As a result of this integration authors were able to establish acquisition of ISA in their previous research in which is the major focus of the introduction and importance of g-SGFET. However, limited neurophysiological analysis was performed here around ISA itself and its relevance to other electrophysiological activity or functions. The authors mostly focused on higher frequencies such as gamma oscillations and high voltage spindles that can be observed and analyzed using traditional multi-electrode arrays. It is certainly encouraging to be able to observe similar activity to what current electrodes can provide but considering the strong emphasis of the technology and its advantages in ultra-low frequency regime I think it would be important to investigate ISA events. I understand that discovery science is not the focus of the manuscript and it is not reasonable to request a discovery while the authors provided extensive yield, biocompatibility and integration results. However, perhaps simple analysis such as demonstration of ISA fluctuation over sleep states or changes of ISA power during physiological events not previously characterized in the infra-slow regime would dramatically strengthen the manuscript, making their technology more attractive and applicable.

We thank the reviewer for this suggestion.

Indeed, demonstration of the capability of the new methodology was the goal of the analysis shown in Figures 5 and 6, that demonstrate utility of the long-term chronic untethered recording in combination with 3D body tracking. Figure 6a,b,d,h demonstrates most novel and interesting result related to ISA: topographic region-specific ISA profiles associated with high voltage spindles (HVS) that are comparable during both immobility and REM sleep (see Fig. 6d in the main text). Here ISA appears in the form of DC fluctuations (with distinct topography-dependent polarity and amplitude) coinciding with HVS course.

To directly address the suggestion of the reviewer, we have evaluated the signal power in the ISA band (0.05-0.15) across brain states, and, most strikingly, found significant increase of ISA power during REM sleep compared to SWS, readily visible during the transition from SWS to REM. Following up on physiological role of the ISA dynamics we found that the phase of ISA modulates power of hippocampal theta oscillation (detected due to volume conduction in posterior cortical regions) and cortical gamma oscillations during REM sleep and power of sleep spindles during SWS. These results are all novel and represent first time demonstration of modulation of known brain rhythms by the phase of infraslow DC rhythm in behaving animals. We now plan a follow-up study with large number

of animals to replicate and investigate these findings at depth. We thank the reviewer for the idea that stimulated this analysis and significantly strengthened the technology validation part of the manuscript.

These new analysis are now added to the results section and figures in the main text and supporting information. We hope the extended results fit the type of demonstration of attractiveness and applicability of the methodology to unravelling infra-slow multichannel extracellular LFP/EEG patterns in chronic freely-moving recordings.

“Classification of brain states is typically based on the delta, alpha-beta and theta frequency bands (see Methods section), reflecting fast-time scale state-specific network dynamics. However, some recent research highlighted the role of infra-slow dynamics in the regulation of brain sub-states⁵⁰, via modulation of higher LFP frequency bands during sleep^{51–53} and dynamic coordination and segregation of the resting state^{54,55}. These results show the potential importance of ISA for a complete classification and study of brain-states. The graphene-based recording system presented here represents an ideal tool for the study of cortical ISA signals with a high accuracy and spatial resolution in freely behaving animals. The spectrogram in Fig. 5b illustrates changes of the spectral power for frequencies between 0.015 and 4 Hz over the transition between SWS and REM. It is possible to observe clear increase in the ISA-band power following the transition from SWS to REM, even at the single trial level (see Fig. 5b). Taking advantage of the long-term recording capabilities of our system, we could sample 44 of such sparsely occurring SWS-REM (REM duration longer than 40 seconds) state transitions within a 24h period. Besides, the spatial mapping of ISA enabled by the g-SGFET technology allows to resolve the topographic region-specific modulation of ISA at the SWS-REM state transition (see supporting information S12). Interestingly, delta-band power, associated with slow oscillations, and infra-slow power showed changes in opposite directions between SWS and REM sleep. While delta band power expectedly decreases from SWS to REM, associated with desynchronized cortical state, infra-slow power increases in REM (see Fig. 5d,e, Fig. S12 and statistical analysis in Methods).

In order to further illustrate the wide frequency band sensitivity of the recording system, we quantified the strength of modulation of LFP power in the slow frequency range (1-15 Hz) by the phase of the ISA activity during REM and SWS. Interestingly, ISA phase significantly modulated theta power (8-9 Hz) during REM sleep (Fig. 5f-left) and spindle band power (9-13 Hz) during SWS (Fig. 5f-right). The strength of ISA phase modulation was ten-fold higher during REM compared to SWS, and the ISA phase of maximal LFP power differed between states being close to the peak ($\sim 340^\circ$) in REM and ascending phase ($\sim 300^\circ$) in SWS. Taking advantage of the large coverage of the cortical mantle by our array, we assessed the spatial extent of the ISA phase modulation of LFP power across cortex, with both theta power during REM and spindle power during SWS showing strongest modulation in posterior part of the array (Fig 5g). While theta oscillations measured on the cortical surface are generated by volume conduction of multiple theta-rhythmic current generators of entorhino-hippocampal circuits^{56,57}, sleep spindles are generated by rhythmic currents of thalamo-cortical projections to granular cortical layers⁵⁸. The fact that power of hippocampal theta and cortical spindle band is modulated by the phase of ISA derived from cortical surface likely reflects global infra-slow dynamics that co-modulates both limbic and cortical circuits. While topographic profile of theta power (Fig. 5g) modulation by ISA phase is consistent with anatomical localization of underlying hippocampal theta current generators, stronger modulation of the spindle power on posterior cortical areas might reflect anatomical thalamo-cortical subcircuits that are more strongly co-modulated by ISA dynamics than derived from epicortical DC signal. Finally, we tested whether g-SGFETs SNR is sufficient to detect fluctuations in the high frequency LFP dynamics at different time scales and to this end quantified the strength of modulation of broad range gamma power (30-200 Hz) by both ISA phase and theta rhythm phase during REM sleep. Gamma power in the range of 60-120 Hz was modulated by the ISA phase reaching maximum power at the peak of the ISA ($\sim 10^\circ$) (Fig. 5h-left) and, consistently to published work based on intracranial recordings⁵⁸, high gamma (120-150 Hz) power was modulated by theta phase (Fig. 5h-right).

Figure 5 | Infra-slow to high-gamma band correlates of sleep and behavioral states. **a**, 3D trajectories of the head position of the rat. The inset shows a scheme of the position of the Mocap cameras with respect to the maze. **b**, The spectrogram of an illustrative channel is displayed together with representative raw LFP signal segments for distinct brain states (top); slow-wave (SW), high-voltage spindles (HVS) and Theta. Movement speed is displayed along with classification of motor state (middle). Brain state is derived from integrating spectral features of LFP with motor state (bottom). **c**, The percentage of time in the active vs inactive state for each hour in a ~22h long recording (i.e. ~24h minus the interruptions to replace the battery) is plotted in the top panel. The percentage of time the rat was in each main brain state is displayed in the bottom panel. **d**, Average 0.015-4 Hz spectrogram for one DC channel triggered on REM episode onsets (n=44). Note opposite dynamics of ISA and slow oscillation (1-4 Hz) band around REM onset. **e**, Median PSD across all SWS-REM transition episodes for 30 second periods pre and post REM onset. Shaded area marks frequency bins with significant difference of PSD between pre and post (p<0.05, permutation test). **f**, Color-coded strength of modulation of LFP power across slow frequency range (y-axis) for one ECoG channel by the phase of ISA across 0.05-0.2 Hz range derived from one DC channel (see panel g) during REM sleep (left) and SWS (right) states; gray color – nonsignificant; insets – circular plot of LFP power in theta / spindle band as a function of ISA phase. **g**, Color-coded topographic maps of ISA phase modulation of LFP power in theta band during REM (left) and spindle band during SWS (right). ISA phase derived from DC channel marked with a red square. **h**, Color-coded strength of modulation of LFP power across gamma frequency range (y-axis) for one ECoG channel by the phase of ISA across 0.05-0.2 Hz range derived from one DC channel (left) and by the phase of LFP in the slow frequency range rhythm (right) during REM. Inset, circular plot

of LFP gamma power with respect to respective (ISA or theta) phase. **i**, Average spectrogram for high frequency range of LFP on posterior channel triggered on the rear onset. Note decrease of 90-200Hz high frequency power and increase of 50-70Hz gamma band power following rear onset. **j**, Head elevation (left) and high gamma power (right) color-coded and centered on rearing onset shown for all events sorted by duration of rear event. Note a clear reduction of high-gamma for the duration of elevated head position. “

We have added the following paragraph in the *Discussion and Outlook* section:

“In the low frequency range, we found that infra-slow power <0.2 Hz increased significantly across DC sites during REM sleep episodes compared to SWS and thus showed the opposite state dependence than power in the slow oscillation band (1-4 Hz). Interestingly, infra-slow dynamics modulated power of theta and gamma rhythm during REM and with lower strength, power of sleep spindles during SWS. While modulation of LFP power in theta, beta and gamma bands by ISA phase derived from BOLD signal and DC EEG has been shown in humans^{59,60} and recently in anesthetized rats⁵⁵, the present result is the first demonstration of interaction between physiologically established oscillatory dynamics, theta, spindle and gamma oscillations, and ECoG-derived ISA in freely-moving rodent. Consistent with published intracranial work⁵⁸, volume conducted hippocampal theta measured at the cortical surface also modulated cortical gamma power, thus demonstrating that the developed technology is sufficiently sensitive to characterize known gamma dynamics.”

We have included a more detailed discussion of the statistical analysis in the Methods section *Statistical analysis*:

“The modulation of ISA power by REM state vs SWS was evaluated in two ways: first, for the period directly around the state transition (-30 to 30s around the REM onset). Secondly, we evaluated the ISA power in both states over their entire duration, not only in the SWS-REM transition. ISA power comparison between REM vs SWS states was restricted to the 44 REM episodes lasting more than 40 seconds (see Supplementary Information S12B). To test for frequency specific changes across the state transition we compared the distributions of median spectral power across trials for the 30 seconds pre vs post SWS-REM transition for each frequency bin. Significance was assessed by permutation test for each frequency bin (n=1000 permutations, see Fig. 5d example channel and Supplementary Information S12C for all working DC channels). Increase of ISA and concurrent decrease of 1-4 Hz power during SWS-REM transition is significant after permutation test (n=1000 permutations) on all DC channels except one excluded channel, same as for longitudinal evaluation in Fig. 3a and 3e, due to poor signal to noise (Supplementary Information S12C). Secondly, we tested for statistical differences between distributions of the integrated power in the ISA (0.01-0.1 Hz) band in SWS and REM states using Wilcoxon rank sum test (see Fig 5e, test results for each channel in Supplementary Information S12D).

Modulation of the LFP power by the DC signal-derived ISA and LFP phase was quantified using instantaneous fast frequency power-weighted resultant length of the instantaneous slow frequency phase vectors normalized by mean LFP power in the respective band⁵⁷ for which magnitude reflects the strength of LFP power modulation to a preferred phase of the ISA or LFP. ISA and LFP phase and LFP power were computed as angle and absolute value of the analytical signal of the respective AC and DC channel signals that were band-pass filtered, with 0.04 and 0.4 Hz bandwidth, respectively. Significance of the modulation was tested based on 1000 surrogate phase-power pairs randomly shifted with respect to each other by up to 100 seconds. Resulting empirical p-value was corrected following false-discovery-rate control procedure at the error rate of 0.001. For constructing topographic maps of theta and spindle power band modulation we used mean modulation strength for the LFP power band 8-9 Hz (theta) and 10-14 Hz (spindle band) and the ISA phase frequency of 0.05-0.1 Hz computed for every AC channel (LFP power) and one fronto-medial DC channel (ISA phase).

And the following supporting data in the *Supplementary Information* section:

“S12. Slow and infra-slow power in the SWS-REM transition.

S12. Infra-slow vs slow oscillation band power during SWS/REM sleep: A, Average spectrogram triggered on the REM onsets for 7 DC channels with high signal to noise. B, Head speed in the SWS to REM transition for all detected events (left). On the right plot the same data is shown for a shorter time window, where the REM events with shorter duration than 40s can be identified. C, Median PSD computed for the 30s pre and post REM onsets. Colored area indicates statistically significant difference (permutation test n=1000 permutations, p<0.05). D, Kernel distribution estimates for the integrated power in 0.01-0.1Hz band during SWS and REM state. p-value for the Wilcoxon ranksum test are indicated.

The authors' effort in providing detailed statistical analysis, population data and reproducibility of the data is certainly appreciated. Such studies have the potential to improve our understanding of brain function and create unique interdisciplinary knowledge.

We thank the reviewer for this comment.

Reviewer #2 (Remarks to the Author):

The authors report on the use of graphene transistors that demonstrate long-term and wireless mapping of wide-bandwidth brain activity which together with a detailed noise analysis are the main findings of the manuscript. The manuscript is very interesting and demonstrates the possible applications of transistors in neurophysiology beside bioelectronic applications of OECTs. The authors, who characterized the noise behavior of graphene transistors in great detail, have clearly worked out the advantages of these electronic components over conventional microelectrode systems in electrophysiological recordings.

We thank the reviewer for this positive evaluation of our work.

Minor points:

Results

Homogeneity and sensitivity of graphene active sensor technology:

- which frequency bandwidth was used for V_{gs-rms} measurements

The bandwidth for the data in Fig. 2b and 2d is from 1-10Hz (indicated in the figure caption). Besides, in Fig. 2g, the V_{gs-rms} noise is presented for different frequency bands (0.05-0.5Hz, 1-10Hz and 20-200Hz) to demonstrate the effect of amplification and digitation noise on the wide-band sensitivity of the system.

- the authors have excluded outliers in the gate equivalent noise representation from their analysis. Is this correct? What is the reason?

In Fig 2 all the outliers are presented. In the boxplot in Fig. 2a,b they are represented by dots. In Fig. 2c,d they are included in the histogram, but they were not considered to calculate the gaussian and log-normal fitting, as indicated in the figure caption. The reason to exclude them from the fitting is that outliers are expected to present anomalous noise, not representative for the statistical properties of intrinsic flicker noise variability in g-SGFETs. However, all the data is represented in the figure.

- it is difficult to verify the low rms noise of 4 uV in Fig 2d taken the wide distribution of in Fig. 2b and I would expect rather an average of 10 -15 uV for V_{gs-rms}

In the first place we would like to note that the V_{gs-rms} distribution plotted in Fig. 2d corresponds to a particular neural probe (device #3 in Fig. 2b).

Secondly, we want to thank the reviewer because indeed there was a mismatch between the distribution shown in Fig. 2b and 2d, which was due to a mismatch in the integration bandwidth. The RMS noise plotted in Fig. 2b corresponds to the 1-100Hz band, while the distribution in Fig. 2d was calculated in the 1-10Hz band. We have corrected Fig. 2b to match the 1-10Hz bandwidth specified in the figure caption. Now it is possible to validate the correspondence between the data plotted in Fig. 2b and 2d.

- Fig 2 h is misleading: it looks like the time domain signals measured in the DC mode are much smaller than those measured in the AC mode. I expect rather an opposite behavior from the 1/f noise

This may be indeed a counter-intuitive result, thus we decided to include it in the main text to clarify

its interpretation. The RMS noise, or variance of the 1/f noise signal, is equal in different frequency bands as long as the integration bandwidth is the same (in a logarithmic scale):

$$I_{ds-rms} = \sqrt{\int_{f_{min}}^{f_{max}} a/f df} = \sqrt{a \ln\left(\frac{f_{max}}{f_{min}}\right)}$$

where a is the current noise power at 1Hz. This equivalence is shown in the probability density plotted in Fig. 2h. However, given a certain variance of the signal, the occurrence of fluctuations for higher and lower frequency bands present different time-scales. Therefore, the number of low frequency fluctuations per unit time is much lower in low-frequency bands, creating the impression of a lower noise. We have added a comment in the section "Wireless headstage design and characteristics" to clarify this interpretation.

"The histogram plotted next to the time-domain representation of both signals shows their probability density distribution, which demonstrates the similarity of their variance, **as expected from the integration of a 1/f spectrum in these frequency bands. Note that the apparently lower amplitude in the time-domain representation of the infra-slow noise is due to the different timescales of 1/f noise in both frequency bands, but not due to a different signal variance.**"

Signal stability and sensitivity over time:

- Fig. 3a shows the shift in the CNP over a time period of 4 weeks. Was a shift towards more neg V_{GS} also observed or only a shift towards more pos values? How much is the shift of the reference electrodes during this time period?

The shift of the CNP was always towards lower values (a negative shift) as represented in Fig. 3a. The contributions from adsorption/desorption of charged chemical species on the graphene channel and changes in the reference potential cannot be distinguished from the experiments reported as indicated in the main text. We have slightly modified the corresponding section in order to clarify this issue.

"The observed shift in the CNP is presumably due to a combination of factors including desorption of contaminants by electrochemical cleaning of the graphene-electrolyte interface⁶¹, adsorption of charged chemical species present in the environment or changes in the reference electrode potential (see supplementary information S3). **However, from these results it is not possible to distinguish among all different contributions.**"

Reviewer #3 (Remarks to the Author):

The authors have developed and tested a 64-ch flexible electrode array incorporating flexible graphene transistors. Graphene is a promising material for future neural interface technologies, particularly for its use in multiplexed recording systems that enable the number of electrodes to scale without increasing the number of external wiring connections. In this work, the authors present the yield and performance uniformity of graphene active sensor arrays fabricated in their cleanroom. They also present chronic, in vivo, wireless recording for 30 days after implant and histology confirming preliminary biocompatibility of the arrays after 12 weeks. The work is important and well done. Congratulations on a very good paper.

We would like to thank the reviewer for the congratulations.

I have a few questions and small edits for the authors to address.

Questions:

1. Can you confirm the size of the electrode / graphene gate / neural sensing area? Is it 100 μm x 100 μm ? That wasn't clear to me.
2. Related, you mention that graphene active arrays are able to better record infraslow brain signals (<0.5 Hz), but I don't see any data to confirm this. In our experience, 200 μm diameter PtIr electrodes have low enough impedance and stability that makes them capable of recording ultra-slow signals, down to 0.005 Hz, with very low noise. Can you compare the in vitro noise

levels (<0.5 Hz) of the graphene active devices to comparably sized passive electrodes? Perhaps using Pt or PtIr or Pt Black? Are there any other factors that make graphene active arrays more capable of recording ISA than traditional faradaic passive electrodes?

The area of the g-SGFET was indeed $100\ \mu\text{m} \times 100\ \mu\text{m}$, it is indicated in the section "*Homogeneity and sensitivity of graphene active sensor technology*" in the main text.

Regarding the sensitivity of the system to detect infra-slow activity (<0.5 Hz), in Fig. 2f we provide the wide-band (0.01-500Hz) noise PSD of the system evaluated *in-vitro*. The RMS noise in the infra-slow frequency band (0.05-0.5Hz) was calculated and plotted in Fig. 2g for the 8 DC-coupled channels of the system, demonstrating an RMS noise dominated by the intrinsic noise of g-SGFET, and therefore not affected by the amplification electronics. Physiological validation of the infra-slow recording capability can be considered as a benchmark test. In the newly added results we show that lowest power ISA is associated with slow-wave sleep and SNR in the ISA band is sufficient to detect significant modulation of the sleep spindle power by the ISA signal phase. Thus, our g-SGFET methodology is capable of capturing and resolving phase of the weakest infra-slow signals across brain states.

For a comparison with electrodes made of other materials (Au and Pt Black) we kindly refer the reviewer to the recent article by Masvidal et.al.³. In that work the fidelity of infra-slow signals detected using g-SGFET was compared with $50\ \mu\text{m}$ diameter Au and Pt Black electrodes and an Ag/AgCl wire in a solution-filled glass micropipette. These results showed the advantage of g-SGFETs with respect to passive electrodes, which presented comparably larger potential drifts and were prone to signal distortion due to the gain drop (and phase shift) at low frequencies^{3,62}. When scaling down the dimensions of passive electrodes, the gain drop and phase shift are expected to play a greater role⁶², decreasing the sensitivity and reliability of the system.

In the present study we have quantified for the first time the noise in the infra-slow band for graphene-based active sensors from a system perspective. Our results show that the sensitivity of g-SGFETs in the infra-slow frequency band is equal to their intrinsic sensitivity in higher-frequency LFP bands for areas smaller than $100\ \mu\text{m} \times 100\ \mu\text{m}$. The relatively large dimensions of the graphene active sensors used in our study demonstrate an upper bound below which the sensitivity of the g-SGFET in the ISA band is not affected by the amplification electronics. Graphene active sensors with smaller dimensions are expected to present a higher intrinsic 1/f noise, as described in Fig.S2 and as expected for any passive or active sensor, while their transconductance and the noise from the transimpedance amplifiers remain unperturbed. Therefore, the relative contribution from the amplification electronics to the low frequency noise of the system is expected to be lower for smaller g-SGFET dimensions. This is in strong contrast with ISA detection using passive electrodes, for which the gain loss and signal distortion is expected to increase for smaller sensor dimensions. Therefore, our results demonstrate the limits and the scalability of the g-SGFET technology towards higher density arrays with ISA detection capabilities from a system perspective. We have included part of this discussion in the section "Wireless headstage design and characteristics".

"Smaller g-SGFETs are expected to present a higher intrinsic noise (see supplementary information Figure S2), as expected for any active or passive sensor. Therefore, our results indicate that the sensitivity of g-SGFETs in the infra-slow frequency band is not affected by the amplification electronics for sensor areas below $100\ \mu\text{m} \times 100\ \mu\text{m}$. This is in strong contrast with ISA detection using passive electrodes, for which the gain loss and signal distortion is expected to increase for smaller sensor dimensions⁶². Therefore, our results demonstrate the limits and the scalability of the g-SGFET technology towards higher density arrays with ISA detection capabilities."

3. Line 183 - Are you using the full scale of the ADC with the gain provided? It seems like it would only use a small part of the ADC scale, given the red curve in figure 1b. A 1mV p-p signal should yield approximately 100 nA signal, yielding a 12 mV signal after IV conversion of 12,000 and a voltage gain of 10. Is that correct?

The calculation of the reviewer is correct. Indeed, we are not filling the full scale of the ADC for the AC channels. It might be possible to further increase the gain of the second amplification stage to further mitigate the effect of digitation noise. However, we expect the floor noise of the

transimpedance amplifier to dominate if the gain is increased excessively. Alternatively, one could slightly increase the drain-to-source current or the W/L ratio of the transistors, but that would also increase the dynamic range in the first amplification stage. We have included a comment in the section "Wireless headstage design and characteristics".

"The digitation noise for AC-channels might be decreased by further optimizing the gain of the second amplification stage. However, the intrinsic noise of the amplifier is expected to dominate for large amplification gains."

4. Supplementary materials S2 - S2. Intrinsic 1/f noise and headstage noise, line 68. I thought this should be 1/A not 1/sqrt(a) because the formula for flicker noise is:
 $V_2 / \text{Hz} = K / (\text{Cox} * W * L * f)$

Indeed the 1/f noise power is typically inversely proportional to the area. However, in the Fig.S2 we represent the equivalent RMS noise at the gate which is calculated as $\sqrt{\int_{f_{min}}^{f_{max}} S_{V_{gs}} df}$, thus the $1/\sqrt{A}$ dependence.

All lines refer to main text unless otherwise stated

Small edits

1. Line 29 - "maturity", there may be a better way to say this or define what you mean.

We believe that technological "maturity" is typically used to designate the current performance and reliability of a technology. However, we agree that its meaning can be clarified in the main text. We have included the following comment in the introduction:

"In this article, we present a sensing system composed of a flexible 64-channel g-SGFET array and a wireless headstage (Fig. 1c-f, and supplementary information S1), which we use to demonstrate **the maturity of this technology in terms of** long-term and wide frequency-band recording capabilities in freely moving animals **from a system perspective**. First, the focus is placed on [...]"

2. Line 64 - Should say "curves at a particular Vds" rather than "curves over Vds"

Thank you for this correction. Here, we referred to the normalized G_m in [S/V] (i.e. $G_m/V_{ds} = g_m$). Thus, we implied "the slope of the I-V curves divided by Vds". We have replaced "over" by "divided by".

3. Line 180 - Should say fig 2E instead of 2D

4. Line 182 - Should say fig 2E instead of 2D

5. Line 185 - Should say fig 2E instead of 2D

6. Line 202 - Should say Fig 2H instead of 3H

Thank you for the corrections, we have implemented the changes.

7. Figure 1f - would be good to include spacing and size of electrode sites

Agree, we have included these geometrical factors.

8. Line 338 - why did you choose these particular cytokines?

The cytokines analysed by ELISA were chosen based on their ability to provide information on the inflammatory state of the brain. After brain injury, such as stroke or trauma, pro-inflammatory TNF-alpha levels can be increased, involved in glial signalling⁶³. IFN-γ is released peripherally by immune cells and can enter the brain if there is blood-brain barrier damage, where it can have pro-inflammatory effects on cells. IL-6 can have both protective or damaging effects on neurons, so

changes in the levels of IL-6 could indicate an imbalance in the normal function of cells in the area. Finally, IL-17a is another pro-inflammatory cytokine, which can be produced by astrocytes, and is recently thought to underlie neurodegeneration as a result of its effects on neurons⁶⁴. All four cytokines are widely regarded as key markers of neuro-inflammation.

9. Line 450 - Should this be LFP instead of LPF?

10. Line 453 - small error but this should be Figure 5

Correct, thank you.

References

1. Kostarelos, K., Vincent, M., Hebert, C. & Garrido, J. A. Graphene in the Design and Engineering of Next-Generation Neural Interfaces. *Adv. Mater.* **29**, 1700909 (2017).
2. Hébert, C. *et al.* Flexible Graphene Solution-Gated Field-Effect Transistors: Efficient Transducers for Micro-Electrocorticography. *Adv. Funct. Mater.* 1703976 (2017). doi:10.1002/adfm.201703976
3. Masvidal-Codina, E. *et al.* High-resolution mapping of infraslow cortical brain activity enabled by graphene microtransistors. *Nat. Mater.* **18**, 280–288 (2019).
4. Garcia-Cortadella, R. *et al.* Switchless Multiplexing of Graphene Active Sensor Arrays for Brain Mapping. *Nano Lett.* acs.nanolett.0c00467 (2020). doi:10.1021/acs.nanolett.0c00467
5. Khodagholy, D. *et al.* NeuroGrid: Recording action potentials from the surface of the brain. *Nat. Neurosci.* **18**, 310–315 (2015).
6. Lee, W. *et al.* Integration of Organic Electrochemical and Field-Effect Transistors for Ultraflexible, High Temporal Resolution Electrophysiology Arrays. *Adv. Mater.* **28**, 9722–9728 (2016).
7. Khodagholy, D. *et al.* *Organic electronics for high-resolution electrocorticography of the human brain.*
8. Chiang, C.-H. *et al.* Development of a neural interface for high-definition, long-term recording in rodents and nonhuman primates. *Sci. Transl. Med.* **12**, eaay4682 (2020).
9. Chung, J. E. *et al.* High-Density, Long-Lasting, and Multi-region Electrophysiological Recordings Using Polymer Electrode Arrays. *Neuron* **101**, 21-31.e5 (2019).
10. Schaefer, N. *et al.* Multiplexed neural sensor array of graphene solution-gated field-effect transistors. *2D Mater.* **7**, 025046 (2020).
11. Viventi, J. *et al.* Flexible, foldable, actively multiplexed, high-density electrode array for mapping brain activity in vivo. *Nat. Neurosci.* **14**, 1599–1605 (2011).
12. Fang, H. *et al.* Capacitively coupled arrays of multiplexed flexible silicon transistors for long-term cardiac electrophysiology. *Nat. Biomed. Eng.* **1**, 0038 (2017).
13. Chang, E. F. Towards large-scale, human-based, mesoscopic neurotechnologies. *Neuron* **86**, 68–78 (2015).

14. Lacour, S. P., Courtine, G. & Guck, J. Materials and technologies for soft implantable neuroprostheses. *Nature Reviews Materials* **1**, 1–14 (2016).
15. Nguyen, J. K. *et al.* Mechanically-compliant intracortical implants reduce the neuroinflammatory response. *J. Neural Eng.* **11**, 056014 (2014).
16. Moshayedi, P. *et al.* The relationship between glial cell mechanosensitivity and foreign body reactions in the central nervous system. *Biomaterials* **35**, 3919–3925 (2014).
17. Song, E., Li, J., Won, S. M., Bai, W. & Rogers, J. A. Materials for flexible bioelectronic systems as chronic neural interfaces. *Nature Materials* **19**, 590–603 (2020).
18. Jastrzebska-Perfect, P. *et al.* Translational Neuroelectronics. *Adv. Funct. Mater.* (2020). doi:10.1002/adfm.201909165
19. Cea, C. *et al.* Enhancement-mode ion-based transistor as a comprehensive interface and real-time processing unit for in vivo electrophysiology. *Nat. Mater.* **19**, 679–686 (2020).
20. Khodagholy, D. *et al.* In vivo recordings of brain activity using organic transistors. *Nat. Commun.* **4**, 1575 (2013).
21. Rivnay, J. *et al.* Organic electrochemical transistors. *Nature Reviews Materials* **3**, 1–14 (2018).
22. Dankerl, M. *et al.* Diamond Transistor Array for Extracellular Recording From Electrogenic Cells. *Adv. Funct. Mater.* **19**, 2915–2923 (2009).
23. Fromherz, P., Offenhäusser, A., Vetter, T. & Weis, J. A neuron-silicon junction: A Retzius cell of the leech on an insulated-gate field-effect transistor. *Science (80-.)*. **252**, 1290–1293 (1991).
24. Spyropoulos, G. D., Gelinas, J. N. & Khodagholy, D. Internal ion-gated organic electrochemical transistor: A building block for integrated bioelectronics. *Sci. Adv.* **5**, eaau7378 (2020).
25. Lee, C., Wei, X., Kysar, J. W. & Hone, J. Measurement of the elastic properties and intrinsic strength of monolayer graphene. *Science (80-.)*. **321**, 385–388 (2008).
26. Zandiatashbar, A. *et al.* Effect of defects on the intrinsic strength and stiffness of graphene. *Nat. Commun.* **5**, 1–9 (2014).
27. Banhart, F., Kotakoski, J. & Krasheninnikov, A. V. Structural Defects in Graphene. *ACS Nano* **5**, 26–41 (2011).
28. Bendali, A. *et al.* Purified Neurons can Survive on Peptide-Free Graphene Layers. *Adv. Healthc. Mater.* **2**, 929–933 (2013).
29. Bullock, C. J. & Bussy, C. Biocompatibility Considerations in the Design of Graphene Biomedical Materials. *Adv. Mater. Interfaces* 1900229 (2019). doi:10.1002/admi.201900229
30. Balasubramanian, K. *et al.* Reversible defect engineering in graphene grain

- boundaries. *Nat. Commun.* **10**, (2019).
31. Banszerus, L. *et al.* Ultrahigh-mobility graphene devices from chemical vapor deposition on reusable copper. *Sci. Adv.* **1**, e1500222 (2015).
 32. Mitra, A. *et al.* Spontaneous Infra-slow Brain Activity Has Unique Spatiotemporal Dynamics and Laminar Structure. *Neuron* **98**, 297-305.e6 (2018).
 33. Pan, W.-J., Thompson, G. J., Magnuson, M. E., Jaeger, D. & Keilholz, S. Infralow LFP correlates to resting-state fMRI BOLD signals. *Neuroimage* **74**, 288 (2013).
 34. Krishnan, G. P., González, O. C. & Bazhenov, M. Origin of slow spontaneous resting-state neuronal fluctuations in brain networks. doi:10.1073/pnas.1715841115
 35. Garcia-Cortadella, R. *et al.* Distortion-Free Sensing of Neural Activity Using Graphene Transistors. *Small* 1906640 (2020). doi:10.1002/sml.201906640
 36. Berto, M. *et al.* EGOFET Peptide Aptasensor for Label-Free Detection of Inflammatory Cytokines in Complex Fluids. *Adv. Biosyst.* **2**, 1700072 (2018).
 37. Khodagholy, D. *et al.* High transconductance organic electrochemical transistors. *Nat. Commun.* **4**, 1–6 (2013).
 38. Mavredakis, N., Garcia Cortadella, R., Bonaccini Calia, A., Garrido, J. A. & Jiménez, D. Understanding the bias dependence of low frequency noise in single layer graphene FETs. *Nanoscale* **10**, 14947–14956 (2018).
 39. Schaefer, N. *et al.* Improved metal-graphene contacts for low-noise, high-density microtransistor arrays for neural sensing. *Carbon N. Y.* **161**, 647–655 (2020).
 40. Hassler, C., Boretius, T. & Stieglitz, T. Polymers for neural implants. *J. Polym. Sci. Part B Polym. Phys.* **49**, 18–33 (2011).
 41. Lu, H. *et al.* SU8-based micro neural probe for enhanced chronic in-vivo recording of spike signals from regenerated axons. in *Proceedings of IEEE Sensors* 66–69 (2006). doi:10.1109/ICSENS.2007.355719
 42. Márton, G. *et al.* A multimodal, SU-8 - Platinum - Polyimide microelectrode array for chronic in vivo neurophysiology. *PLoS One* **10**, (2015).
 43. Hugo, A. Capteurs biologiques à base de transistors graphène à grille liquide. <http://www.theses.fr> (2020).
 44. Rubehn, B. & Stieglitz, T. In vitro evaluation of the long-term stability of polyimide as a material for neural implants. *Biomaterials* **31**, 3449–3458 (2010).
 45. Hess, L. H. *et al.* Graphene Transistor Arrays for Recording Action Potentials from Electrogenic Cells. *Adv. Mater.* **23**, 5045–5049 (2011).
 46. Guzinski, M. *et al.* PEDOT(PSS) as Solid Contact for Ion-Selective Electrodes: The Influence of the PEDOT(PSS) Film Thickness on the Equilibration Times. *Anal. Chem.* **89**, 3508–3516 (2017).
 47. Mackin, C. *et al.* A Current–Voltage Model for Graphene Electrolyte-Gated Field-Effect Transistors. *IEEE Trans. Electron Devices* **61**, 3971–3977 (2014).

48. Rivnay, J. *et al.* High-performance transistors for bioelectronics through tuning of channel thickness. *Sci. Adv.* **1**, e1400251 (2015).
49. Masvidal-Codina, E. *et al.* High-resolution mapping of infraslow cortical brain activity enabled by graphene microtransistors. *Nat. Mater.* **18**, 280–288 (2019).
50. Watson, B. O., Hengen, K. B., Gonzalez Andino, S. L. & Thompson, G. J. Cognitive and Physiologic Impacts of the Infraslow Oscillation. *Front. Syst. Neurosci* **12**, 44 (2018).
51. Vanhatalo, S. *et al.* Infraslow oscillations modulate excitability and interictal epileptic activity in the human cortex during sleep. *Proc. Natl. Acad. Sci.* **101**, 5053–5057 (2004).
52. Leopold, D. A. Very Slow Activity Fluctuations in Monkey Visual Cortex: Implications for Functional Brain Imaging. *Cereb. Cortex* **13**, 422–433 (2003).
53. Lecci, S. *et al.* Coordinated infraslow neural and cardiac oscillations mark fragility and offline periods in mammalian sleep. *Sci. Adv.* **3**, e1602026 (2017).
54. Hiltunen, T. *et al.* Infra-Slow EEG Fluctuations Are Correlated with Resting-State Network Dynamics in fMRI. *J. Neurosci.* **34**, 356–362 (2014).
55. Thompson, G. J. *et al.* Phase-amplitude coupling and infraslow (<1 Hz) frequencies in the rat brain: Relationship to resting state fMRI. *Front. Integr. Neurosci.* **8**, 41 (2014).
56. Sirota, A. *et al.* Entrainment of Neocortical Neurons and Gamma Oscillations by the Hippocampal Theta Rhythm. *Neuron* **60**, 683–697 (2008).
57. Pesaran, B. *et al.* Investigating large-scale brain dynamics using field potential recordings: Analysis and interpretation. *Nat. Neurosci.* **21**, 903–919 (2018).
58. Sirota, A. & Buzsáki, G. Interaction between neocortical and hippocampal networks via slow oscillations. *Thalamus Relat. Syst.* **3**, 245–259 (2007).
59. Pan, W. J., Thompson, G. J., Magnuson, M. E., Jaeger, D. & Keilholz, S. Infraslow LFP correlates to resting-state fMRI BOLD signals. *Neuroimage* **74**, 288–297 (2013).
60. Grooms, J. K. *et al.* Infraslow Electroencephalographic and Dynamic Resting State Network Activity. *Brain Connect.* **7**, 265–280 (2017).
61. Fu, W. *et al.* Biosensing near the neutrality point of graphene. *Sci. Adv.* **3**, e1701247 (2017).
62. Nelson, M. J., Pouget, P., Nilsen, E. A., Patten, C. D. & Schall, J. D. Review of signal distortion through metal microelectrode recording circuits and filters. *J. Neurosci. Methods* **169**, 141–157 (2008).
63. Dugue, R., Nath, M., Dugue, A. & Barone, F. C. Roles of Pro- and Anti-inflammatory Cytokines in Traumatic Brain Injury and Acute Ischemic Stroke. in *Mechanisms of Neuroinflammation* (InTech, 2017). doi:10.5772/intechopen.70099
64. Zimmermann, J. *et al.* CNS-Targeted Production of IL-17A Induces Glial Activation, Microvascular Pathology and Enhances the Neuroinflammatory Response to Systemic Endotoxemia. *PLoS One* **8**, 57307 (2013).

REVIEWERS' COMMENTS

Reviewer #1 (Remarks to the Author):

In this revised manuscript, authors substantially improved their manuscript from the previous version. I have no further comments and I am impressed by their research and manuscript.

Reviewer #2 (Remarks to the Author):

The authors have addressed all my comments and I can recommend the publication of the manuscript in the current version

Reviewer #3 (Remarks to the Author):

The authors have addressed my concerns.

Some additional notes:

- SU8 is known to fail rapidly, particularly when biased in an active electrode. I'd recommend changing to a more robust encapsulation material as soon as possible.
- Make sure you include a leakage current measurement channel in your recording system to be able to detect when the encapsulation has failed and shut down the array power. Otherwise, you will expose the animal to continuous DC currents that are damaging to the brain and will cause seizures.

REVIEWERS' COMMENTS

Reviewer #1 (Remarks to the Author):

In this revised manuscript, authors substantially improved their manuscript from the previous version. I have no further comments and I am impressed by their research and manuscript.

Thank you for your detailed review. It really helped improve our manuscript.

Reviewer #2 (Remarks to the Author):

The authors have addressed all my comments and I can recommend the publication of the manuscript in the current version

Thank you for the positive, yet rigorous review.

Reviewer #3 (Remarks to the Author):

The authors have addressed my concerns.

Some additional notes:

- SU8 is known to fail rapidly, particularly when biased in an active electrode. I'd recommend changing to a more robust encapsulation material as soon as possible.
- Make sure you include a leakage current measurement channel in your recording system to be able to detect when the encapsulation has failed and shut down the array power. Otherwise, you will expose the animal to continuous DC currents that are damaging to the brain and will cause seizures.

Thank you for your review and for this latest suggestion. Indeed, this is a critical aspect for intracranial probes. We have made an important effort in the last months to improve the encapsulation of devices, and we are confident we will be able to report on alternative passivation procedures in the near future. Regarding control of DC leakage, monitoring of the leakage current in-vivo would be a nice add-on. We are also working on an alternative operation mode to limit the current leakage in case of device failure. We hope we will report it in the near future as well.

Thank you again